# Single-Teacher View Augmentation: Boosting Knowledge Distillation via Angular Diversity

**Seonghoon Yu**[1,*]    **Dongjun Nam**[2,*]    **Dina Katabi**[3]    **Jeany Son**[2]

[1]GIST         [2]POSTECH         [3]MIT CSAIL

seonghoon@gm.gist.ac.kr   dina@csail.mit.edu   {june6423,jeany}@postech.ac.kr

https://github.com/june6423/Angular-KD

## Abstract

Knowledge Distillation (KD) aims to train a lightweight student model by transferring knowledge from a large, high-capacity teacher. Recent studies have shown that leveraging diverse teacher perspectives can significantly improve distillation performance; however, achieving such diversity typically requires multiple teacher networks, leading to high computational costs. In this work, we propose a novel cost-efficient knowledge augmentation method for KD that generates diverse multi-views by attaching multiple branches to a single teacher. To ensure meaningful semantic variation across multi-views, we introduce two angular diversity objectives: 1) *constrained inter-angle diversify loss*, which maximizes angles between augmented views while preserving proximity to the original teacher output, and 2) *intra-angle diversify loss*, which encourages an even distribution of views around the original output. The ensembled knowledge from these angularly diverse views, along with the original teacher, is distilled into the student. We further theoretically demonstrate that our objectives increase the diversity among ensemble members and thereby reduce the upper bound of the ensemble's expected loss, leading to more effective distillation. Experimental results show that our method surpasses an existing knowledge augmentation method across diverse configurations. Moreover, the proposed method is compatible with other KD frameworks in a plug-and-play fashion, providing consistent improvements in generalization performance.

## 1 Introduction

Knowledge Distillation (KD) has emerged as a powerful paradigm for model compression, aiming to transfer knowledge from a large, high-capacity teacher network to a compact student model. By guiding the student not only through hard labels but also via richer supervisory signals, such as softened class probabilities [19, 62, 22], intermediate features [6, 5, 15, 39], or attention maps [58], KD enables the student to learn nuanced decision boundaries and fine-grained representations. This approach significantly improves the performance of lightweight models, making it highly suitable for deployment in resource-constrained environments, such as mobile devices [7], embedded systems [63], and IoT platforms [1].

Building upon this foundation, recent studies have introduced multi-teacher distillation [3, 20, 45, 36], where the student learns from the collective knowledge of multiple teachers, offering a richer and more diverse supervisory signal. By aggregating complementary insights from distinct teacher models, these approaches significantly enhance the student's generalization capabilities. However, the benefits of multi-teacher frameworks come with increased computational and memory costs due to the need to train and maintain multiple large models. Moreover, the diversity in such frameworks often arises from initializing identical model architectures with different random seeds (as in ensemble distillation [3]), which inherently limits diversity due to shared structural biases [38].

---

*Equal contributions

39th Conference on Neural Information Processing Systems (NeurIPS 2025).

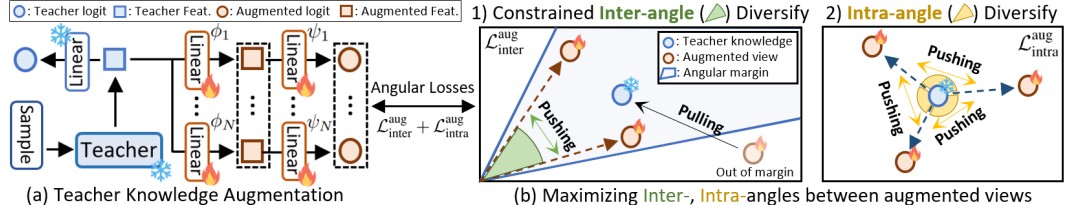

Figure 1: The illustration of our angularly diverse knowledge augmentation: (a) Multi-views are generated from a single pre-trained teacher using multiple pathways, and (b) these augmented outputs are then optimized with two complementary angular objectives to maximize their intra- and inter-angular diversity.

Recently, TeKAP [20] was introduced as a knowledge augmentation strategy that simulates multi-perspective supervision by adding multiple stochastic perturbations into features in a single teacher model. This approach reduces the computational cost of multi-teacher distillation by avoiding the need to train multiple teacher networks. However, the diversity it induces is entirely driven by random noise, offering limited control over the semantic structure or informativeness of the augmented outputs. This raises the question of whether more structured and controllable augmentation can lead to more effective knowledge transfer, which is a direction we explore in this work.

In this work, we propose a lightweight knowledge augmentation approach that generates diverse perspectives from a single teacher by attaching multiple pathways (Fig. 1a), thereby eliminating the overhead of training multiple teacher networks. Unlike a noise-based method [20], our approach enables explicit control over the diversity of augmented views while preserving semantic consistency. To ensure attached linear branches produce complementary and informative knowledge, we introduce two novel learning objectives (Fig. 1b) that maximize angular diversity among the augmented outputs. First, the *constrained **inter-angle** diversity loss* maximizes the angular separation among all augmented views while constraining them within a margin around the original teacher output to maintain alignment with the source knowledge. Second, the ***intra-angle** diversity loss* encourages an even spread of the augmented views centered on the teacher's original view, promoting broad coverage of the local knowledge space. Together, these objectives generate a set of semantically rich and angularly diverse teacher views. These views are aggregated with the original teacher output to provide a stronger and more informative supervisory signal for student distillation. In addition, it significantly reduces computational cost by eliminating the need for training multiple teacher models.

Furthermore, we theoretically demonstrate that our proposed objectives increase ensemble diversity [38], which in turn tightens the upper bound on the expected loss, an effect directly linked to improved distillation performance [34]. Empirically, our approach achieves state-of-the-art performance on standard knowledge distillation benchmarks under various teacher-student configurations. Moreover, it consistently enhances student generalization when integrated as a plug-and-play module into existing distillation frameworks.

Our contributions can be summarized as follows:

- We propose Angular-KD, a novel knowledge augmentation framework that generates multiple diverse views from a single teacher by attaching lightweight, learnable linear branches. This eliminates the need to train and store multi-teacher models.

- To explicitly control and enhance the diversity of the augmented views, we introduce two angular objectives: 1) a *constrained **inter-angle** diversity loss* that encourages angular separation while preserving alignment with the original teacher prediction, and 2) an ***intra-angle** diversity loss* that enforces a uniform spread around the teacher output.

- We provide both theoretical analysis and empirical evidence that the proposed angular diversity objectives lead to more diverse and structured augmented views, which consistently improve student performance across a wide range of distillation settings.

## 2   Method

In this section, we present Angular-KD, a lightweight yet effective knowledge augmentation framework that generates diverse supervisory signals from a single pre-trained teacher. The core idea is to

append multiple lightweight linear branches to the teacher, each generating distinct logits or features that serve as diverse views for distillation (Fig. 1a). To ensure these augmented outputs provide complementary supervision, we optimize them by maximizing both inter- and intra-angular diversity, while constraining their output within the target class boundaries (Fig. 1b). This encourages the generation of rich knowledge without incurring the costs of training multiple teacher models.

We begin by describing the detailed design of *single-teacher view augmentation heads*, which generate multiple augmented views from the original teacher knowledge (Sec. 2.1). We then introduce the *inter- and intra-angular diversity objectives*, which explicitly optimize these views to achieve the desired angular diversity (Sec. 2.2). Finally, we describe how the student is distilled with both the augmented and original teacher knowledge in an end-to-end manner (Sec. 2.3).

## 2.1 Single-teacher View Augmentation Heads

To generate diverse views of the teacher's knowledge using a single network, we attach multiple linear transformation heads, referred to as view augmentation heads, to the teacher. To encourage structural diversity at initialization, each head is initialized using orthogonal initialization [43], allowing them to explore distinct representational directions. This reduces redundancy among learned features and leads to more diverse and informative representations early in training. Additionally, we apply dropout [46] with varying probabilities to the input features of each head, introducing stochastic variation and further encouraging semantic diversity among the views. The architectural design of the view augmentation heads is described in detail below.

**Teacher Knowledge Extraction.** We utilize two forms of teacher knowledge: the final feature representation and the logit probability. To enable multi-view augmentation, we first extract the teacher knowledge at both feature and logit levels using the teacher network. Given an input sample $\mathbf{x}$, the teacher network first produces the final feature vector $\mathbf{F}^T = f(\mathbf{x}; \theta^T) \in \mathbb{R}^{d_T}$, where $f(\cdot; \theta^T)$ denotes the teacher's feature extractor, parameterized by $\theta^T$, and $d_T$ is the dimensionality of the feature representation. The corresponding logit probabilities $\mathbf{Z}^T$ are then computed by applying a classification layer $\mathbf{W}^T \in \mathbb{R}^{C \times d_T}$ to the extracted features: $\mathbf{Z}^T = \sigma(\mathbf{W}^T \mathbf{F}^T / \tau^Z) \in \mathbb{R}^C$, where $\sigma(\cdot)$ is the softmax function, $C$ is the number of classes, and $\tau^Z$ is a temperature scaling factor to control the smoothness of the distribution.

**View Augmentation Heads.** To generate $N$ diverse views from the extracted teacher knowledge, we attach a set of $N$ linear transformation heads to the teacher. These view augmentation heads are independently applied at the feature and logit levels, producing alternative representations from a single teacher source. We first attach $N$ number of multiple *feature-level augmentation heads* $\{\phi_i\}_{i=1}^N$ with orthogonal initialization [43] to promote diversity across views. To further encourage semantic variation in views, we apply a dropout mask $M_i$ with dropout probability $m_i$ to the input feature before passing it through each head. The $i$-th augmented feature is then computed as:

$$\mathbf{F}_i^A = \phi_i(\mathbf{F}^T), \quad \text{where } \phi_i(\mathbf{F}^T) = \text{BN}(\mathbf{W}^{\phi_i}(M_i \odot \mathbf{F}^T)) \in \mathbb{R}^{d_T}, \quad i \in \{1, \ldots, N\}, \quad (1)$$

where $\text{BN}(\cdot)$ is a BatchNorm layer, $\mathbf{W}^{\phi_i} \in \mathbb{R}^{d_T \times d_T}$ is a linear layer, and $\odot$ denotes the element-wise product. Augmented features $\mathbf{F}_i^A$ are derived from a unique pair of dropout masks $M_i$ and feature-level heads $\phi_i$, enabling the generation of diverse views for distillation.

Each augmented feature is further transformed into a logit probability $\mathbf{Z}_i^A \in \mathbb{R}^C$ over $C$ classes. This is achieved by passing the augmented feature $\mathbf{F}_i^A$ through a *logit-level augmentation head* $\psi_i$, consisting of a linear layer $\mathbf{W}^{\psi_i} \in \mathbb{R}^{C \times d_T}$, followed by a softmax $\sigma(\cdot)$ with a temperature $\tau^Z$:

$$\mathbf{Z}_i^A = \psi_i(\mathbf{F}_i^A), \quad \text{where } \psi_i(\mathbf{F}_i^A) = \sigma(\mathbf{W}^{\psi_i} \mathbf{F}_i^A / \tau^Z) \in \mathbb{R}^C, \quad i \in \{1, \ldots, N\}. \quad (2)$$

The resulting augmented features $\{\mathbf{F}_i^A\}_{i=1}^N$ and logits $\{\mathbf{Z}_i^A\}_{i=1}^N$ are jointly optimized to maximize angular diversity by updating the parameters of the view augmentation heads $\{\phi_i\}_{i=1}^N$ and $\{\psi_i\}_{i=1}^N$ using our proposed angular diversity loss functions (see Sec. 2.2). These augmented view pairs $\{\mathbf{F}_i^A, \mathbf{Z}_i^A\}_{i=1}^N$ are also distilled into the student model alongside the original teacher pair $\{\mathbf{F}^T, \mathbf{Z}^T\}$, as detailed in Sec. 2.3.

## 2.2 Angular Losses for Learning Diverse Representations

While orthogonal initialization and dropout introduce initial diversity in the view augmentation heads, they are insufficient to ensure meaningful separation between the generated views. To explicitly

encourage representational diversity across views during training, we introduce two angular diversity loss functions: *Constrained inter-angle diversity loss* and *Intra-angle diversity loss*.

**Constrained Inter-angle Diversity Loss.** The constrained inter-angular diversity loss $\mathcal{L}_{\text{inter}}^{\text{aug}}$ is designed to maximize the angular separation among the augmented knowledge by minimizing their cosine similarity, thereby capturing the absolute deviation in representation space (Fig. 1b). We adopt cosine similarity due to its numerical stability compared to directly optimizing the arccosine [11]. However, excessive angular deviation may cause augmented views to drift toward non-target class boundaries. To mitigate this, we introduce a constraint that encourages each augmented view to remain within a learnable angular margin from the teacher's view. Once all augmented views satisfy this constraint, a diversity term is activated to further maximize inter-view separation. Formally,

$$
\mathcal{L}_{\text{inter}}^{\text{aug}} = \underbrace{-\sum_{i=1}^{N} \log \frac{\exp\left(\min(1, \gamma + s_i^T)/\tau^C\right)}{\sum_{k \in \mathcal{N}_i} \exp\left(s_k^T/\tau^C\right)}}_{\text{Constraint term}} + \underbrace{\mathbb{1}_{\left\{\forall i \in \{1,\ldots,N\}: \gamma + s_i^T \geq 1\right\}} \sum_{i=1}^{N} \sum_{j \neq i}^{N} s_{ij}^A}_{\text{Diversity term}} \quad (3)
$$

$$
\text{where} \quad s_i^T = \begin{cases} \cos(\mathbf{F}^T, \mathbf{F}_i^A) & \text{(feature-level)} \\ \cos(\mathbf{Z}^T, \mathbf{Z}_i^A) & \text{(logit-level)}, \end{cases} \quad s_{ij}^A = \begin{cases} \cos(\mathbf{F}_i^A, \mathbf{F}_j^A) & \text{(feature-level)} \\ \cos(\mathbf{Z}_i^A, \mathbf{Z}_j^A) & \text{(logit-level)}, \end{cases}
$$

$\cos(\mathbf{u}, \mathbf{v}) = \mathbf{u} \cdot \mathbf{v}/\|\mathbf{u}\|\|\mathbf{v}\|$, $\gamma$ is a learnable angular margin, $\tau^C$ is a temperature parameter, $\mathcal{N}_i$ denotes the set of negatives for the $i$-th augmented samples from other batches, and $\mathbb{1}_{\{\cdot\}}$ is an indicator function activated when the specified condition holds.

The loss can be applied either to augmented features $\mathbf{F}_i^A$ or to logits $\mathbf{Z}_i^A$ by computing their angular similarity, encouraging diversity at the representational or predictive level, respectively. To prevent semantic drift, the constraint term penalizes augmented views that fall outside the angular margin, effectively pulling them closer to the teacher representation. Incorporating negatives further encourages discriminative alignment. Once all augmented views reside within the angular margins, the diversity term maximizes inter-view angular discrepancy to enhance diversity across the augmented views.

**Intra-angle Diversity Loss.** While inter-angle loss encourages view separation in the absolute representation space, it does not account for how each view deviates from the teacher. To ensure that the augmented views are evenly distributed around the teacher's knowledge, we introduce the intra-angle diversity loss, $\mathcal{L}_{\text{intra}}^{\text{aug}}$. This objective minimizes the pairwise cosine similarity among the offset vectors $\{\mathbf{\Delta}_i^{T-A}\}_{i=1}^{N}$, which quantify the directional deviation of each augmented view from the teacher representation. Specifically, each offset vector is computed as the difference between the teacher representation and the corresponding augmented instance, i.e., $\mathbf{\Delta}_i^{T\text{-}A} = \mathbf{R}^T - \mathbf{R}_i^A$, where $\mathbf{R} \in \{\mathbf{F}, \mathbf{Z}\}$ denotes either the feature ($\mathbf{F}$) or logit-level ($\mathbf{Z}$) representation. Formally, it is defined as:

$$
\mathcal{L}_{\text{intra}}^{\text{aug}} = \sum_{i=1}^{N} \sum_{j \neq i}^{N} s_{ij}^{\mathbf{\Delta}}, \text{ where } s_{ij}^{\mathbf{\Delta}} = \cos(\mathbf{\Delta}_i^{T-A}, \mathbf{\Delta}_j^{T-A}), \ \mathbf{\Delta}_i^{T-A} = \begin{cases} \mathbf{F}^T - \mathbf{F}_i^A & \text{(feature-level)} \\ \mathbf{Z}^T - \mathbf{Z}_i^A & \text{(logit-level)}. \end{cases} \quad (4)
$$

Minimizing $\mathcal{L}_{\text{intra}}^{\text{aug}}$ encourages structurally diverse offset directions, complementing the inter-angle objective by shaping a more structurally balanced variation around the teacher and promoting diverse, informative knowledge augmentation.

**Overall Loss Function for View Augmentation** In addition to two angular diversity objectives, we supervise each augmented logit with the ground-truth one-hot labels $\mathbf{y} = \{y_1, \ldots, y_c\} \in \mathbb{R}^C$ with $C$ classes, as follows: $\mathcal{L}_{\text{gt}}^{\text{aug}} = \sum_{i=1}^{N} \text{CrossEntropy}(\mathbf{y}, \mathbf{Z}_i^A)$. This supervision acts as a regularization signal to prevent the augmented predictions from diverging too far from the true semantic target. The overall loss function applied to the augmented knowledge is defined as: $\mathcal{L}^{\text{aug}} = \mathcal{L}_{\text{inter}}^{\text{aug}} + \mathcal{L}_{\text{intra}}^{\text{aug}} + \mathcal{L}_{\text{gt}}^{\text{aug}}$. We refer to *feature-level augmentation* as the setting where the angular diversity loss $\mathcal{L}^{\text{aug}}$ is applied exclusively to the outputs of the feature-level augmentation heads, while *logit-level augmentation* applies the loss solely to the outputs of the logit-level heads. Our full model incorporates both *logit- and feature-level augmentation*, enforcing angular diversity at both the feature- and logit-level representations. This encourages consistency and diversity across both the representational and predictive spaces, leading to more robust and generalizable learning.

## 2.3 Distillation with Augmented Knowledge

To distill the augmented teacher knowledge into the student, we construct a $(N+1)$-way ensemble by averaging the original teacher output with its $N$ augmented variants. The resulting ensemble representation is denoted as $\mathbf{F}^E$ or $\mathbf{Z}^E$, depending on whether feature- or logit-level outputs are used. This combined representation serves as supervision for the student model. For feature-level distillation, we employ a Contrastive Representation Distillation (CRD) [50] loss, $\mathcal{L}_{\text{feat.}}^{\text{distill}}$, to align the student's features with the ensembled teacher representation. For logit-level distillation, we employ a Kullback-Leibler (KL) divergence loss [19], $\mathcal{L}_{\text{logit}}^{\text{distill}} = \text{KL}(\mathbf{Z}^E, \mathbf{Z}^S)$, to align the student's logit $\mathbf{Z}^S$ with the ensembled logits aggregated from the teacher and augmented views. Additionally, we apply a standard cross-entropy loss $\mathcal{L}_{\text{gt}}^{\text{distill}} = \text{CrossEntropy}(\mathbf{y}, \mathbf{Z}^S)$ to directly supervise the student with ground-truth labels. The overall distillation loss is defined as $\mathcal{L}^{\text{distill}} = \mathcal{L}_{\text{feat.}}^{\text{distill}} + \mathcal{L}_{\text{logit}}^{\text{distill}} + \mathcal{L}_{\text{gt}}^{\text{distill}}$, where either or both of the feature- and logit-level terms can be included depending on the chosen distillation setting. The augmentation and distillation losses are applied to the teacher and student, respectively, enabling joint end-to-end training of the entire model.

## 3 Theoretical Analysis

To theoretically justify the performance benefits of our approach, we build upon existing analyzes of ensemble diversity [38], which demonstrate that increased diversity among ensemble members reduces the upper bound on the expected ensemble loss. In this section, we formally analyze (1) how our inter- and intra-angle diversity losses relate to the ensemble diversity, and (2) how this amplified diversity tightens the upper bound on the ensemble's expected loss, thereby improving student performance.

**Effect of Intra-, Inter-angular Losses on Ensemble Diversity.** Let $\{\mathbf{Z}_i\}_{i=0}^N$ denote the set of logits used in the ensemble, where $\mathbf{Z}_0 := \mathbf{Z}^T$ is the original teacher output and $\mathbf{Z}_i := \mathbf{Z}_i^A$ are the logits from the $i$-th augmented view for $i = 1, \ldots, N$. The ensemble diversity metric $\mathbb{D}(\cdot)$ [49, 37], defined over the ensemble members $\{\mathbf{Z}_i\}_{i=0}^N$ and dataset $\mathcal{D}$, can be expressed in two equivalent forms, depending on either the pairwise similarity $s_{ij}^A$ (defined in Eq. (3)) between normalized logits or the angular similarity $s_{ij}^{\boldsymbol{\Delta}}$ (defined in Eq. (4)) between offset vectors, as follows:

$$\mathbb{D}(\{\mathbf{Z}_i\}_{i=0}^N) = \mathbb{E}_{(\mathbf{x},\mathbf{y})\sim\mathcal{D}}\left[\mathbb{V}_{i=0}^N\left[\frac{\mathbf{Z}_i}{\max \mathbf{Z}_i}\right]\right] = \begin{cases} \mathbb{E}_{(\mathbf{x},\mathbf{y})\sim D}\left[\mathbb{E}_{i=1}^N\left[\|\mathbf{Z}_i\|^2\right] - \mathbb{E}_{i,j=1}^N\left[\|\mathbf{Z}_i\|\|\mathbf{Z}_j\|s_{ij}^A\right]\right], & (5\text{a}) \\ -\mathbb{E}_{(\mathbf{x},\mathbf{y})\sim D}\left[\mathbb{E}_{i,j=1}^N\left[\|\boldsymbol{\Delta}_i^{T-A}\|\|\boldsymbol{\Delta}_j^{T-A}\|s_{ij}^{\boldsymbol{\Delta}}\right]\right]. & (5\text{b}) \end{cases}$$

The full derivation is provided in the supplementary material. These two equations indicate that ensemble diversity $\mathbb{D}$ grows as the cosine similarities $s_{ij}^A$ and $s_{ij}^{\boldsymbol{\Delta}}$ decrease. Notably, the inter-angle diversity loss (Eq. (3)) explicitly minimizes $s_{ij}^A$, while the intra-angle diversity loss (Eq. (4)) targets $s_{ij}^{\boldsymbol{\Delta}}$. Together, these losses reduce the similarity terms in the ensemble diversity metric $\mathbb{D}$, thereby increasing diversity and tightening the upper bound on the ensemble's expected loss, as shown below.

**Upper Bound on Expected Loss of Ensemble.** Similar to the analysis in [38], the expected loss $\mathbb{E}_{(\mathbf{x},\mathbf{y})\sim\mathcal{D}}[\mathcal{L}(\cdot)]$ of the ensemble logit $\mathbf{Z}^E$ over the dataset $\mathcal{D}$ is bounded by

$$\mathbb{E}_{(\mathbf{x},\mathbf{y})\sim\mathcal{D}}\left[\mathcal{L}(\mathbf{Z}^E)\right] \leq \mathbb{E}_{(\mathbf{x},\mathbf{y})\sim\mathcal{D}}\left[\mathbb{E}_{i=0}^N\left[\mathcal{L}(\mathbf{Z}_i)\right]\right] - K\mathbb{D}(\{\mathbf{Z}_i\}_{i=0}^N), \quad (6)$$

where $\mathcal{L}$ is a cross-entropy loss and $K > 0$ is a constant. This bound shows that increasing the ensemble diversity $\mathbb{D}$ lowers the upper bound on the ensemble's expected loss. Furthermore, this bound is also tightened by an improvement in the average quality of the individual views. By maximizing angular diversity, our method lowers this bound, strengthening the ensemble supervision and improving the student performance [34]. The detailed derivation is also included in our supplementary.

## 4 Experiments

### 4.1 Experimental Setting

**Dataset and Metric.** We conduct experiments on various KD benchmark datasets: **1) CIFAR-100** [24], a 100-class image classification dataset containing 50,000 training and 10,000 validation images of size $32 \times 32$, **2) ImageNet** [10]: a large-scale classification dataset with 1,000 categories, approximately 1.28 million training and 50,000 validation images, each of size $224 \times 224$, and **3)**

Table 1: **Results on CIFAR-100.** Top-1 test accuracy (%) of student models across various teacher-student configurations compared with a SoTA augmentation method [20]. Augmentations are applied at three different levels: *logit-level augmentation* with logit distillation (KD), *feature-level augmentation* with feature distillation (CRD), and *logit- and feature-level augmentation* with logit and feature distillation (KD + CRD).

| Arch. of Teacher / Student | Same Architecture | | | Different Architecture | | |
|---|---|---|---|---|---|---|
| | RN32×4 | VGG13 | WideRN-40-2 | RN32×4 | RN32×4 | WideRN-40-2 |
| | RN8×4 | VGG8 | WideRN-40-1 | WideRN-40-2 | WideRN-16-2 | RN8×4 |
| Acc. of Teacher | 79.42 | 74.64 | 75.61 | 79.42 | 79.42 | 75.61 |
| Acc. of Student | 72.50 | 70.36 | 71.98 | 75.61 | 73.26 | 72.50 |
| *Logit-level augmentation* | | | | | | |
| Logit Distil. [19] w/o aug | 73.33 | 72.98 | 73.54 | 77.70 | 74.70 | 73.97 |
| w/ TeKAP [20] | 74.79 | 74.00 | 73.80 | 77.97 | 75.08 | 75.09 |
| w/ Angular-KD (ours) | **76.08** | **74.57** | **74.86** | **78.68** | **76.22** | **76.27** |
| *Feature-level augmentation* | | | | | | |
| Feat. Distil. [50] w/o aug | 75.51 | 73.94 | 74.14 | 78.15 | 75.65 | 75.24 |
| w/ TeKAP [20] | 75.65 | 74.10 | 74.21 | 78.05 | 75.42 | 75.65 |
| w/ Angular-KD (ours) | **75.82** | **74.55** | **74.91** | **78.51** | **76.19** | **76.20** |
| *Logit- and feature-level augmentation* | | | | | | |
| Combined Distil. w/o aug | 75.46 | 74.29 | 74.38 | 78.01 | 75.86 | 75.57 |
| w/ TeKAP [20] | 75.98 | 74.42 | 74.41 | 78.68 | 75.62 | 75.90 |
| w/ Angular-KD (ours) | **76.46** | **74.76** | **75.00** | **78.94** | **76.42** | **76.29** |

Table 2: **Plug-and-Play Results on CIFAR-100**. Comparison with a SoTA augmentation method [20] for KD. Three scenarios are reported: the original SoTA KD method without any augmentation, augmented with TeKAP, and augmented with our Angular-KD.

| Arch. of Teacher / Student | Same Architecture | | | Different Architecture | | |
|---|---|---|---|---|---|---|
| | RN32×4 | VGG13 | WideRN-40-2 | RN32×4 | RN32×4 | WideRN-40-2 |
| | RN8×4 | VGG8 | WideRN-40-1 | WideRN-40-2 | WideRN-16-2 | RN8×4 |
| Acc. of Teacher | 79.42 | 74.64 | 75.61 | 79.42 | 79.42 | 75.61 |
| Acc. of Student | 72.50 | 70.36 | 71.98 | 75.61 | 73.26 | 72.50 |
| DKD [62] w/o aug | 76.32 | 74.68 | 74.81 | 78.46 | 75.70 | 75.56 |
| w/ TeKAP [20] | **76.65** | 74.55 | 73.83 | 78.64 | 75.28 | **76.22** |
| w/ Angular-KD (ours) | 76.51 | **74.76** | **74.89** | **78.99** | **76.05** | 76.14 |
| ReviewKD [6] w/o aug | 75.63 | 74.10 | 75.09 | 78.96 | 76.11 | 74.34 |
| w/ TeKAP [20] | 75.32 | **74.29** | 75.15 | **79.55** | 76.36 | 75.59 |
| w/ Angular-KD (ours) | **75.78** | 74.12 | **75.45** | 78.28 | **76.40** | **75.97** |
| MLKD [22] w/o aug | 77.08 | 75.18 | 75.35 | 79.26 | 76.52 | 77.33 |
| w/ TeKAP [20] | 77.04 | 75.37 | 75.31 | 78.72 | 76.46 | 77.28 |
| w/ Angular-KD (ours) | **77.28** | **75.63** | **75.37** | **79.52** | **76.60** | **77.45** |

**Imbalanced CIFAR-100**, following prior works [50, 20], where 43 classes out of 100 CIFAR classes are selected, and each class is limited to 50 training samples. The imbalanced classes can be found in our supplementary. **4) STL-10** [8], a 10-class image classification dataset with an image size of $96 \times 96$, comprising 5,000 training images and 8,000 test images. **5) TinyImageNet** [10], a 200-class subset of ImageNet, with each class containing 500 training images, 50 validation images, and 50 test images with a size of $64 \times 64$. For the evaluation metric, we use top-1 classification accuracy (%).

**Implementation Details.** For view augmentation heads (Sec. 2.1), we generate $N = 5$ augmented views, apply dropout with probabilities $\{0.2, 0.25, 0.3, 0.35, 0.4\}$, and use a softmax temperature of $\tau^Z = 4$. For constrained inter-angle diversity loss (Sec. 2.2), the learnable margin $\gamma$ is initialized to 0.2, and the contrastive temperature is set to $\tau^C = 0.07$. For the ensembling (Sec. 2.3), we use uniform weights across all ensemble members. Training starts with a 30-epoch warm-up phase where only the view augmentation heads are trained to ensure stability. Subsequently, the student model is trained for 240 epochs on a single RTX 2080 Ti GPU using SGD optimizer, and a batch size of 64. The learning rate starts at 0.01 and is decayed by a factor of 10 at epochs 150, 180, and 210.

## 4.2 Main Result

**Results on CIFAR-100.** Table 1 reports a comparison between the proposed Angular-KD and and the prior augmentation method [20], on CIFAR-100. We evaluate three augmentation levels: (1)

Table 3: **Results on ImageNet.** The logit-level augmentations are applied for both Angular-KD and TeKAP under a ResNet34 teacher and a ResNet18 student.

| Metric | Teacher | Student | w/o aug | TeKAP | Ours |
|---|---|---|---|---|---|
| Top-1 Acc. | 73.31 | 69.75 | 70.41 | 70.67 | **71.07** |
| Top-5 Acc. | 91.42 | 89.07 | 89.88 | 89.92 | **90.39** |

Table 4: **Results on Binary Segmentation** on the Carbana Image Masking dataset.

| Methods | Dice Loss ↓ | IoU |
|---|---|---|
| Naïve KD | 0.0218 | 94.83 |
| + Ours | **0.0208** | **95.72** |

Table 5: **Results on imbalanced CIFAR-100.** "*Imbalanced set*" includes 43 under-sampled classes, "*Full set*" indicates a total 100 classes with extra balanced 57 categories.

| Class type | w/o aug | TeKAP | Ours |
|---|---|---|---|
| Imbalanced set | 33.23 | 34.98 | **35.66** |
| Full set | 60.74 | 61.18 | **61.84** |

Table 6: **Transferability Results.** The distilled knowledge on CIFAR-100 is transferred to STL-10 and TinyImageNet. The classifier for each dataset is trained on top of a frozen student.

| Dataset | w/o aug | TeKAP | Ours |
|---|---|---|---|
| STL-10 | 68.01 | 68.71 | **70.23** |
| TinyImageNet | 31.17 | 31.54 | **32.97** |

logit-level augmentation with logit distillation (KD [19]), (2) feature-level augmentation with feature distillation (CRD [50]), and (3) their combination. Evaluations are conducted across multiple teacher-student configurations under two scenarios: *same-architecture*, where both teacher and student use the same network, and *cross-architecture*, where different backbones are employed. In the table, we denote ResNet [17] and WideResNet [57] as *RN* and *WideRN*, respectively. Across all configurations, Angular-KD consistently outperforms TeKAP by substantial margins, highlighting the effectiveness of angular-diversity-based augmentations over TeKAP's random noise perturbations.

**Plug-and-Play Results on CIFAR-100.** Table 2 presents the distillation performance of our Angular-KD when integrated as a plug-and-play module on top of SoTA KD methods [62, 22, 6], compared to TeKAP [20], under both *same architecture* and *different architecture* settings. For most methods, adding Angular-KD yields consistent performance gains, and it outperforms TeKAP in most settings, demonstrating its effectiveness and generalization in generating valuable supervisory signals for the student through our angularly diverse augmentation.

**Results on ImageNet.** To validate the scalability, we evaluate Angular-KD on the ImageNet [10] validation set using a ResNet34 teacher and a ResNet18 student. Table 3 reports Top-1 and Top-5 accuracies (%) for both Angular-KD and TeKAP under logit-level augmentations with naïve KD [19] Our method achieves the best results compared to the unaugmented one and TeKAP, confirming its robustness on large-scale datasets. The detailed implementations are presented in the supplementary.

**Results on Binary Segmentation.** To further validate that our approach is effective beyond the image classification task, we conduct a binary segmentation task on the Carvana Image Masking dataset [44], which includes 5,088 training images and 100,064 test images. Using a U-Net-32 teacher and a U-Net-16 student, we compare the augmented KD with our Angular-KD against the unaugmented KD baseline. As shown in Table 4, our approach achieves substantial improvements over the baseline across two metrics (Dice loss [35] and IoU), demonstrating its task generality beyond image classification.

**Results on Imbalanced CIFAR-100.** We assess robustness on an imbalanced CIFAR-100 dataset, where 43 of the 100 classes are undersampled to 50 training images (See supplementary for details). The student is assessed on two test settings: (1) the 47-class imbalanced subset and (2) the full 100-class set including 53 balanced classes. All models use a WideRN-40-2 teacher and a WideRN-16-2 student, with both feature- and logit-level augmentations. As shown in Table 5, Angular-KD achieves the highest accuracy in both settings, highlighting strong robustness in imbalanced scenarios.

**Transferability Results on STL-10 and TinyImageNet.** To validate that our Angular-KD leads a student to a more general model, we evaluate the CIFAR-100 distilled student (via unaugmented, TeKAP, or Angular-KD) on STL-10 and TinyImageNet. For this experiment, we train a classifier for each transferred dataset on top of the frozen student, and use a WideRN-40-2 teacher and a WideRN-16-2 student. In Table 6, a trained student via Angular-KD achieves the best performance, underscoring that angularly diversified distillation produces a more generalizable representation.

Table 7: **Advantages over multi-teacher approaches.** The best results are highlighted in **bold** and second best are underlined. † indicates Ensemble Distillation [3].

| Metric | WideRN-40-2_Teacher    WideRN-16-2_Student | | | | | VGG13_Teacher    VGG8_Student | | | | |
| | Multi-teacher Methods | | | Single-teacher Aug. | | Multi-teacher Methods | | | Single-teacher Aug. | |
| | ED† [3] | TAKD [36] | DGKD [45] | TeKAP | Ours | ED† [3] | TAKD [36] | DGKD [45] | TeKAP | Ours |
|---|---|---|---|---|---|---|---|---|---|---|
| Acc. | 76.31 | 75.04 | 76.24 | 76.20 | **76.33** | 74.67 | 73.67 | 74.40 | 74.42 | **74.76** |
| Params (M) | 11.28 | 6.69 | 6.69 | **2.26** | 2.40 | 47.31 | 20.31 | 20.31 | **9.46** | 11.04 |
| FLOPs (M) | 1645 | 797 | 797 | **329** | 329 | 1427 | 497 | 497 | **285** | 287 |

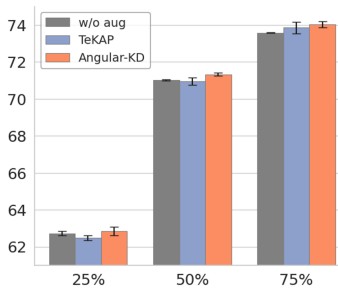

Figure 2: **Few-Shot Results.**

Table 8: **Ablation on each proposed method.** We report accuracy and the ensemble diversity metric (from our theoretical analysis) to assess the impact of each objective.

| Constrained Inter-angle Diversity Loss | Intra-angle Diversity Loss | Acc. | Ensemble Diversity |
|---|---|---|---|
| | | 75.46 | - |
| ✓ | | 76.16 | 11.522 |
| | ✓ | 76.28 | 11.617 |
| ✓ | ✓ | 76.46 | 11.633 |

**Advantages over Multi-teacher Approach.** To highlight our advantages over multi-teacher distillation methods [3, 36, 45], we compare KD accuracy and resource efficiency in Table 7. We consider two configurations: (1) **WideRN setting**, where Angular-KD and TeKAP employ a single WideRN-40-2 teacher and a WideRN-16-2 student, while TAKD [36] and DGKD [45] each train four teachers (WideRN-40-2, -34-2, -28-2, -22-2), and Ensemble Distillation [3] uses five WideRN-40-2 teachers; and (2) **VGG setting**, where Angular-KD and TeKAP use a single VGG13 teacher and a VGG8 student, while TAKD [36] and DGKD [45] train two teachers (VGG13, VGG11), and Ensemble Distillation [3] uses five VGG13 teachers. We report the total number of teacher parameters and FLOPs, which reveal the substantially higher resource cost of multi-teacher methods. Notably, Angular-KD achieves higher accuracy than all baselines while requiring only a single teacher, demonstrating superior efficiency.

**Few-shot Results.** We further evaluate Angular-KD under limited-data scenarios by randomly sampling 25%, 50%, and 75% of the CIFAR-100 training set compared with unaugmented KD + CRD and a SoTA augmentation method, namely TeKAP [20], as shown in Fig. 2. All compared methods use a WideRN-40-2 teacher and a WideRN-16-2 student. Across all sampling ratios, Angular-KD achieves the best average performance over three trials (error bars are shown in the figure). These results indicate that our angular-diversity augmentations provide more informative supervisory signals that help the student generalize even when training data are scarce.

## 4.3 Ablation Study

We conduct an ablation study on CIFAR-100 using ResNet32×4 and ResNet8×4 as the teacher and the student, respectively.

**Effect of Each Proposed Method.** In Table 8, we analyze the contribution of each proposed method in our Angular-KD by reporting accuracy and ensemble diversity (defined in Sec. 3). Beginning with the unaugmented KD + CRD baseline, we then sequentially introduce our angular diversification objectives with augmentation heads. The constrained inter-angle diversity loss alone increases both accuracy and ensemble diversity, indicating that separating views from one another is beneficial. Our intra-angle diversity loss also drives up ensemble diversity, thereby improving performance. When combining both angular-diversity objectives, we achieve the best accuracy and diversity, confirming that inter- and intra-angular objectives work together to strengthen distillation.

**Effect of the Number of Augmentations.** Table 9 explores the effect of varying the number of augmentations ($N$) on performance. Notably, even a single augmentation ($N = 1$) yields improvement over the unaugmented KD + CRD baseline, denoted as "*w/o*" in the table. We obtain

Table 9: **Ablation study on the number of augmentations.** "*w/o aug*" refers to a baseline with logit (KD) and feature (CRD) distillation without augmentation. † denotes teacher's Params and FLOPs.

| Metric | w/o aug | # of Augmentations ($N$) | | | | | |
|---|---|---|---|---|---|---|---|
| | | 1 | 2 | 3 | 4 | **5** | 6 |
| Acc. | 75.46 | 75.87 +0.39 | 75.85 +0.39 | 76.25 +0.79 | 76.44 +0.98 | **76.46** +1.00 | 76.37 +0.91 |
| Params (M) | 7.434† | +0.092 | +0.184 | +0.276 | +0.368 | +0.460 | +0.552 |
| FLOPs (M) | 1085.629† | +0.092 | +0.185 | +0.277 | +0.370 | +0.462 | +0.554 |

Table 10: **Ablation study within view augmentation head.**

| Orth. Init. | Dropout | Acc. |
|---|---|---|
| | | 76.17 |
| ✓ | | 76.31 |
| | ✓ | 76.35 |
| ✓ | ✓ | **76.46** |

Table 11: **Ablation study within the constrained inter-angle diversify loss.**

(a) Constraint, diversity terms

| Constraint | Diversity | Acc. |
|---|---|---|
| ✓ | | 76.25 |
| | ✓ | 76.29 |
| ✓ | ✓ | **76.46** |

(b) Margin $\gamma$

| Margin $\gamma$ | Acc. |
|---|---|
| 0.1 | 76.34 |
| **0.2** | **76.46** |
| 0.3 | 76.31 |

the best results at $N = 5$. Importantly, increasing $N$ incurs marginal computational overhead, as each augmentation adds only a lightweight head with negligible impact, espcially on overall FLOPs.

**Ablation Study within View Augmentation Heads.** We conduct an ablation to isolate the individual effects of orthogonal initialization [43] and input dropout [46] on our view augmentation heads, as shown in Table 10. Initializing each head with orthogonal weights increases accuracy, demonstrating that starting in distinct directions leads to more diverse views. Applying dropout before each head further boosts the results. Combining both orthogonal initialization and dropout yields accuracy gains, confirming that these techniques work synergistically to diversify the augmented knowledge.

**Ablation Study within Constrained Inter-angle Diversity Loss.** We analyze the effects of the constraint and diversity terms, as well as the choice of angular margin $\gamma$ in our constrained inter-angle diversity loss. Table 11a shows that combining both constraint and diversity terms yields the highest accuracy, confirming the effectiveness of our objective design. Table 11b further reveals that setting an angular margin $\gamma$ to 0.2 yields the highest results, offering the best trade-off between promoting diversity and preserving class boundaries.

**t-SNE Visualization.** We visualize the t-SNE embeddings of the original teacher logit and augmented logits produced by our method (Fig. 3a) compared to those from a SoTA augmentation method [20] (Fig. 3b). For clarity, we randomly sample 10 out of the 100 classes and assign a unique color to each category. We observe that our augmented views are more evenly dispersed, particularly for the gray and red classes, while those of TeKAP are clustered. Furthermore, TeKAP's augmented views overlap heavily in the purple and red class regions, unlike ours.

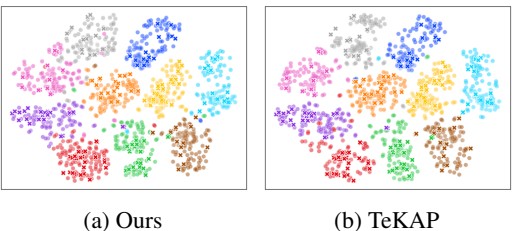

(a) Ours      (b) TeKAP

Figure 3: **The t-SNE visualization** of teacher (✖) and augmented logits (●) from Angular-KD compared to TeKAP.

## 5 Related Work

**Knowledge Distillation.** Knowledge Distillation (KD), first introduced by Hinton et al. [19], train a compact student to mimic the softmax outputs of a large teacher. Since then, 1) logit-based approaches [62, 29, 47, 59, 61, 28] have been developed by modifying logit probabilities, *e.g.* using adaptive temperature [14], or predicting sample-wise temperature [29]. Parallel to this, 2) feature-based approaches [2, 18, 30, 40, 32, 55, 5, 15] transfer intermediate representations directly, such as feature maps [42], and spatial attention maps [58], enabling more fine-grained guidance. To further boost knowledge transfer, multi-teacher distillations [56, 13, 54] aggregate diverse views

from multiple independently trained teachers. While effective, these methods demand significant computational resources. In this work, we replace the array of teachers with lightweight linear branches, drastically reducing the overhead of acquiring diverse knowledge.

**Diversity and Ensemble Learning.** Ensemble methods boost performance by encouraging diversity among their models. Classic Bagging [4] and Boosting [12] do so via resampling or reweighting, while modern deep ensembles use random initializations [27, 52] or stochastic optimization [33, 60, 53]. In contrast, we generate varied predictors through lightweight single-linear branches combined with our angular diversity losses. Furthermore, a range of diversity metrics [37, 23] (*i.e.*, ambiguity [25], orthogonality [26], disagreement [25]) has been proposed. Especially, Ortega et al. [38] formalize a generic diversity measure linked to the generalization error bound. We show that our objectives increase this measure, thereby tightening the theoretical bound and enhancing distillation [34].

**Teacher Knowledge Augmentation.** TeKAP [20] first tackles the high cost of multi-teacher distillation by augmenting single-teacher knowledge into diverse views via random perturbations. While this reduces resource demands, its reliance on undirected, stochastic noise limits the overall diversity. In contrast, our method explicitly maximizes angular separation among generated views via two novel angular-diversity losses, ensuring more complementary, less-redundant augmentations.

**Angle-based Learning Objective.** A series of methods [48, 21, 51] have shown that imposing angular margins on the classifier's hypersphere largely boosts inter-class separation and enforces intra-class compactness. SphereFace [31] replaces softmax logit with $\cos(m\theta)$ to enforce a multiplicative angular margin; CosFace [51] applies an additive margin $m$ via $\cos(\theta)-m$; and ArcFace [11] directly adds an angular offset $\cos(\theta + m)$ to optimize geodesic separability. Unlike these methods, our angular-diversity losses work on multiple augmented views of a single teacher, not on different classes, to harness angular maximization for knowledge diversity rather than class-margin enforcement.

## 6 Discussions

**Limitations.** A key limitation of our approach is that the augmentations are inherently constrained by the original teacher's knowledge, making it difficult to introduce fundamentally new semantic information. While we achieve multi-view diversity through inter- and intra-angular variations, our method does not capture the broader semantic richness that could be offered by multiple distinct teacher models. Moreover, although angular augmentation is computationally more efficient than multi-teacher approaches, generating multiple views still incurs a non-negligible training-time overhead. In future work, we plan to mitigate this limitation by incorporating novel semantic signals beyond those of the original teacher, while keeping additional costs minimal.

**Social Impacts.** Our method, Angular-KD, improves the efficiency and accessibility of knowledge distillation by eliminating the need for multiple large teacher models. This may facilitate the deployment of compact, high-performing models in low-resource environments such as edge devices, mobile platforms, and underserved regions. The reduced computational demand may also contribute to lowering the environmental impact of large-scale training and reducing the overall carbon footprint. However, there are also potential societal risks. As all augmentations are generated from a single teacher, any biases or blind spots present in the teacher model may be transferred or even amplified in the student. This raises concerns related to fairness and representational harm, particularly in sensitive domains such as hiring, finance, healthcare, or surveillance. Furthermore, since angular perturbations are not semantically grounded, the approach may limit interpretability and increase susceptibility to adversarial manipulation.

## 7 Conclusion

In this paper, we present a novel KD augmentation method that generates angularly diverse views from a single teacher using linear layers with two novel angular diversity learning objectives, where valuable findings are provided: (1) Theoretically, angular diversity in augmented views leads to high ensemble diversity, reducing student errors, and (2) empirically, our angularly diverse augmentation achieves the best performance across various KD benchmarks, and delivers additional gains when incorporated as a plug-and-play with existing KD approaches.

**Acknowledgments.** This work was supported by the IITP grants (RS-2019-II191906, RS-2022-II220926) funded by MSIT and the GIST-MIT Research Collaboration grant funded by GIST, Korea.

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

# Single-Teacher View Augmentation: Boosting Knowledge Distillation via Angular Diversity
## *- Supplementary materials -*

## Overview of Supplementary Materials

We provide the table of contents for the supplementary materials below:

## A  Additional Results

### A.1  Additional Plug-and-Play Results with a Multi-teacher Method

In Table 10, we report the KD performance when integrating our Angular-KD into the multi-teacher approach, TAKD [36], compared with when adapting the SoTA KD augmentation method, TeKAP [20]. In this setup, TAKD uses additional teacher assistants (WideRN-22-1 or WideRN-22-2) with the primary teacher network (WideRN-40-2) for multi-teacher distillation. We apply our Angular-KD to all transfer steps: 1) the primary teacher to the assistant, and 2) the assistant to the student. The augmented TAKD with our methods shows superior performance over an unaugmented TAKD and an augmented one with TeKAP, highlighting that introducing angularly diversified views into each teacher assistant yields stronger distillation than random perturbations of TeKAP.

Table 10: **Plug-and-Play Results with TAKD on CIFAR-100**. The WideRN-40-2 teacher is used at all assistant settings.

| Teacher Assistant | WideRN-22-2 | | WideRN-22-1 | |
|---|---|---|---|---|
| **Student** | WideRN-16-2 | WideRN-40-1 | WideRN-16-2 | WideRN-40-1 |
| TAKD [36] | 75.02 | 72.73 | 72.56 | 71.19 |
| w/ TeKAP [20] | 76.02 | 74.02 | 73.02 | 72.47 |
| w/ Angular-KD (ours) | **76.27** | **75.28** | **75.02** | **74.33** |

## A.2 Additional Transferability Results on STL-10 and TinyImageNet

We extend our transferability experiments on both STL-10 and TinyImageNet using 1) a ResNet32×4 teacher and a ResNet8 × 4 student and 2) a VGG13 teacher and a VGG8 student (results for a WideRN-40-2 teacher and a WideRN-16-2 student appear in Table 4 of the main paper). We adopt augmentations at both feature and logit levels. For each method (the unaugmented one, the augmented with TeKAP, and Angular-KD), we freeze the student model distilled from CIFAR-100 and train a linear classifier on each downstream dataset. As shown in Table 11, Angular-KD again achieves the highest accuracy, demonstrating that its benefits carry over to different model backbones, underscoring its broad applicability.

Table 11: **Transferability Results.** The distilled knowledge on CIFAR-100 is transferred to STL-10 and TinyImageNet. The classifier for both datasets is trained on top of a frozen student. The "*WRN*" indicates a WideResNet.

| Dataset | RN32×4$_{Teacher}$   RN8×4$_{Student}$ | | | VGG13$_{Teacher}$   VGG8$_{Student}$ | | | WRN-40-2$_{Teacher}$   WRN-16-2$_{Student}$ | | |
|---|---|---|---|---|---|---|---|---|---|
| | w/o aug | TeKAP | Ours | w/o aug | TeKAP | Ours | w/o aug | TeKAP | Ours |
| STL-10 | 70.66 | 71.70 | **73.25** | 67.11 | 67.53 | **68.43** | 68.01 | 68.71 | **70.23** |
| TinyImageNet | 33.37 | 34.46 | **36.71** | 31.72 | 31.96 | **32.43** | 31.17 | 31.54 | **32.97** |

## A.3 Comparisons on Training Time

To validate the efficiency of our approach, we compare the training time of our methods with 1) naíve KD + CRD without augmentation, 2) the augmented one by TeKAP, and 3) multi-teacher methods (*e.g.*, Ensemble distillation [3]), including computational demands (FLOPs and Params; reported in Table 7) in Table. This experiment is conducted with a WideResNet40-2 teacher and a WideResNet-16-2 student on a 2080 Ti GPU over 50,000 training images and a total of 240 epochs. We observe that our multiple branches incur only a marginal increase in both training time and computation cost (FLOPs and Params), since the branches are composed of lightweight linear heads.

Table 12: **Comparisons on Training Time.** We report training time during 1 epoch over 50,000 CIFAR training images with computational demands (Parameters and FLOPs).

| | w/o aug | TeKAP | Ours | Ensemble Distillation [3] |
|---|---|---|---|---|
| Training Time (s) | 27 | 32 | 35 | 47 |
| Params (M) | 2.26 | 2.26 | 2.40 | 11.28 |
| FLOPs (M) | 329.02 | 329.02 | 329.19 | 1645.10 |

## A.4 Experiments for Statistical Significance

Table 13 presents the results of a 3-trial experiment to evaluate the stability of our methods on CIFAR-100 under each of augmentation level (feature-, logit-, and their combined-levels), using a ResNet32×4 teacher and a ResNet8×4 student (Table 13a) and a ResNet32×4 teacher and a WideResNet40×2 student (Table 13b), compared to TeKAP. We report accuracy for each trial and then compute the average (Avg.) and the standard deviation (Std.) across runs. Our methods consistently outperform TeKAP in average performance across all settings. This consistency indicates that the observed improvements are statistically significant and not a result of random variation.

Table 13: **Experiments for Significal Significance.** We report 3-trial results on CIFAR-100 with the average (Avg.) and the standard deviation (Std.)

| Metric | *Logit-level Distill.* | | | *Feature-level Distill.* | | | *Logit, Feature-level Distill.* | | |
|---|---|---|---|---|---|---|---|---|---|
| | w/o aug | TeKAP | Ours | w/o aug | TeKAP | Ours | w/o aug | TeKAP | Ours |
| Acc. (Top-1) | 73.33 | 75.84$\pm$0.16 | **76.08**$\pm$0.22 | 75.51 | 75.82$\pm$0.15 | **75.94**$\pm$0.11 | 75.46 | 75.89$\pm$0.06 | **76.39**$\pm$0.17 |

(a) A ResNet32×4 teacher and a ResNet8×4 student.

| Metric | *Logit-level Distill.* | | | *Feature-level Distill.* | | | *Logit, Feature-level Distill.* | | |
|---|---|---|---|---|---|---|---|---|---|
| | w/o aug | TeKAP | Ours | w/o aug | TeKAP | Ours | w/o aug | TeKAP | Ours |
| Acc. (Top-1) | 77.70 | 77.99$\pm$0.09 | **78.57**$\pm$0.13 | 78.15 | 77.97$\pm$0.21 | **78.56**$\pm$0.20 | 78.01 | 77.96$\pm$0.30 | **78.62**$\pm$0.05 |

(b) A ResNet32×4 teacher and a WideResNet40×2 student.

# B Analysis on Augmented Views

## B.1 Accuracy for each Augmented View

In Table 14, we report Top-1 accuracy (%) on CIFAR-100 for each augmented view and for the final ensemble between the original teacher and all augmented views, with a WideRN-40-2 teacher and a WideRN-16-2 student. In this experiment, we set $N = 5$ for both our method and TeKAP [20]. Each of our five angularly diversified views performs worse than the original teacher and shows relatively large performance variation. However, when ensembling all five augmented views with the teacher, the overall accuracy improves beyond that of the teacher alone. In contrast, ensembling the teacher with TeKAP's five random perturbations actually lowers performance compared to using the teacher by itself. In addition, our student model also achieves better accuracy than TeKAP. The result shows that our views provide more complementary information, leading to the highest KD accuracy.

Table 14: **Accuracy for each Augmented View.** We report the accuracy of the final ensemble with the original teacher and all augmented views (denoted as "*Ensemble Acc.*"). The ResNet32 $\times$ 4 teacher and a ResNet8 $\times$ 4 student are used. "*Student KD Acc.*" indicates the accuracy of the distilled student model. † denotes the original score reported in the TeKAP paper, while other scores are our reproduced results based on the authors' code.

| Methods | Teacher | Augmented Views $N$ | | | | | Ensemble Acc. | Student KD Acc. |
|---|---|---|---|---|---|---|---|---|
| | | 1 | 2 | 3 | 4 | 5 | | |
| TeKAP [20] | 79.42 | 79.34 | 79.38 | 79.41 | 79.48 | 79.45 | 79.38 | 75.85 / 75.98† |
| Angular-KD (ours) | 79.42 | 79.23 | 79.30 | 79.42 | 79.34 | 79.19 | **79.51** | **76.46** |

## B.2 Effect of Inter- and Inter-angles in Ensemble Diversity, Expected Loss, and Accuracy

In this work, we claim that angular diversity directly leads to improved knowledge distillation, based on the theoretical analysis presented in the main paper. This effect is achieved by (1) increasing ensemble diversity and (2) consequently lowering the upper bound of the ensemble expected loss. To support this key claim, we report the KD performance, average inter- and intra-angle statistics, measured ensemble diversity, and ensemble expected loss between ensembled logits and a ground-truth label in Table 15. As our angular diversity objectives are added, both the mean inter-angle and intra-angle increase. This, in turn, raises the measured ensemble diversity and lowers the measured ensemble's expected loss. Consequently, the KD accuracy improves.

We further compare these results with those of the state-of-the-art KD augmentation method, TeKAP, in Table 16. Notably, TeKAP achieves a higher intra-angle, but all other metrics are inferior to ours. Although TeKAP's random perturbation can boost intra-angle by scattering views randomly around the teacher, it cannot safely scale up noise to increase inter-angle without risking drift into non-target classes. As a result, its inter-angle, ensemble diversity, and expected-loss reductions all fall short of ours. In contrast, our method simultaneously increases both inter- and intra-angle, tightens the ensemble's expected-loss bound, and improves KD accuracy, demonstrating a more robust and effective augmentation strategy.

Table 15: **Effect of Inter- and Intra-angles in Ensemble Diversity and Accuracy** with each component. The reported inter- and intra-angles are averaged value over all augmented views.

| Constrained Inter-angle Diversity Loss | Intra-angle Diversity Loss | Student KD Acc. | Angles | | Ensemble Diversity | Ensemble Loss $\downarrow$ |
|---|---|---|---|---|---|---|
| | | | Inter | Intra | | |
| | | 75.46 | - | - | - | - |
| ✓ | | 76.16 | 11.23° | 11.40° | 11.522 | 0.810 |
| | ✓ | 76.28 | 14.09° | 16.85° | 11.617 | 0.809 |
| ✓ | ✓ | **76.46** | **20.86°** | **25.38°** | **11.633** | **0.806** |

Table 16: **Effect of Inter- and Intra-angles in Ensemble Diversity and Accuracy** compared to TeKAP. The reported inter- and intra-angles are averaged value over all augmented views.

| Methods | Student KD Acc. | Angles | | Ensemble Diversity | Ensemble Loss ↓ |
|---|---|---|---|---|---|
| | | Inter | Intra | | |
| TeKAP [20] | 75.98 | 4.35° | **35.64°** | 6.65 | 0.853 |
| Angular-KD (ours) | **76.46** | **20.86°** | 25.38° | **11.633** | **0.806** |

## B.3 Correlation Matrix between Augmented Views

Fig. 4 shows a comparison of the pairwise cosine similarity matrices (i.e., correlation matrices) between our angularly augmented logits and TeKAP's perturbed logits. For clarity, both methods use the same number of augmentations ($N = 5$). For Angular-KD (Fig. 4a), similarities range widely (*e.g.*, 0.786 between view 1 and view 5), revealing substantial diversity. In contrast, TeKAP's randomly perturbed logits (Fig. 4b) are almost identical, with all pairwise similarities around 0.997. These results demonstrate our angular diversification produces more varied soft logits, providing richer, non-redundant supervision for the student, thereby improving value for distillation.

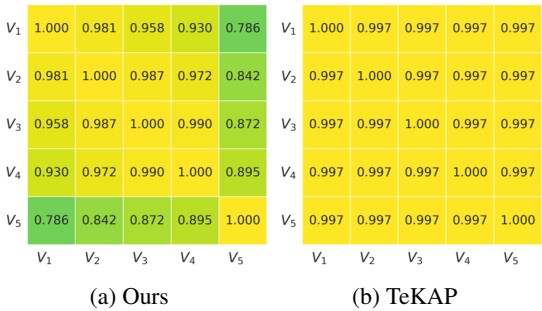

(a) Ours  (b) TeKAP

Figure 4: **Correlation Matrix** between augmented views ($V_1, \ldots, V_5$) compared to TeKAP.

## C  Additional Implementation Details

**ImageNet Experiments.**  We elaborate on the dataset and implementation details for ImageNet experiments, reported in Table 3. **For the dataset**, ImageNet [10] is a large-scale classification dataset with 1,000 categories, approximately 1.28 million training and 50,000 validation images, each of size 224×224. Its large-scale and high-diversity make ImageNet an ideal testbed for evaluating the scalability and robustness of knowledge distillation methods. **For the implementation details,** we first begin with a 10-epoch warm-up training, where only the view-augmentation heads are trained for stability. Then, like previous KD approaches [20, 47], we follow the standard PyTorch training schedule but extend it by 20 additional epochs, decay the learning rate by 0.1× at epochs 40, 70, and 100, and use a batch size of 512.

**Play-and-Play Experiments.**  To integrate our Angular-KD into the SoTA KD methods, *i.e.*, DKD [62] and MLKD [22], we follow their configurations and details of the official code of the corresponding methods. We also perform an initial warm-up training exclusively on the view-augmentation heads during 30 epochs. For the multi-teacher approach, TAKD, we follow a two-stage procedure: first, we train the primary teacher network and distill its knowledge into a single assistant network with augmentations; next, we distill the assistant's knowledge to the student with augmentations.

**Imbalanced CIFAR-100 Experiments.**  To construct the imbalanced CIFAR-100 dataset, we employ the same class selection protocol as TeKAP [20]. In detail, we select 43 classes as imbalanced classes among a total of 100 classes. For each of these classes, we choose the first 50 training samples in sequence (without shuffling) for reproducibility and discard the rest. The remaining 57 classes keep their full set of 500 training images. This yields an imbalanced distribution for evaluating KD methods under severe class imbalance.

**Few-shot Experiments.**  To create the limited-data scenarios, we randomly sample 25%, 50%, and 75% of CIFAR-100 training data. For reproducibility, we select the first 25%, 50%, and 75% samples in sequence (without shuffling). This deterministic selection ensures reproducibility while simulating few-shot learning scenarios.

# D   Data Augmentation *v.s.* Our View Augmentation

Traditional data augmentation methods are commonly used to generate positive views. To evaluate our approach, which generates views via linear pathways, we compare it against the conventional augmentation technique. For experiments, we apply standard data augmentations (random flip, color jiterring, and random rotation) to generate multiple views and ensemble them for student distillation, similar to our view augmentations. Table 17 shows the CIFAR-100 accuracy for our view-augmented logits *v.s.* the data-augmented logits, where a ResNet32×4 teacher and a ResNet8×4 student are used. We observe a substantial performance drop when multiple views are generated via data augmentation. To analyze this degradation, we check the accuracy of each augmented view in Table 18. Each augmented view by data augmentation shows significantly lower accuracy due to the noise caused by the data perturbation, leading to a decline in overall performance. This highlights the effectiveness of our approach in producing consistent and diverse representations without computational overhead.

Table 17: **KD Performance.**

| Methods | Acc. |
|---|---|
| Baseline | 75.46 |
| w/ Data aug. | 66.91 |
| w/ View aug. | **76.46** |

Table 18: **Accuracy for each Augmented View.**

| Methods | Augmented Views $N$ | | | | | Teacher Acc. |
|---|---|---|---|---|---|---|
| | 1 | 2 | 3 | 4 | 5 | |
| Data aug. | 74.52 | 74.62 | 74.56 | 74.41 | 74.30 | 79.42 |
| View aug. | 79.23 | 79.30 | 79.42 | 79.34 | 79.19 | |

Furthermore, our view-augmentation provides two major benefits over the data augmetnation: 1) Efficiency (1 forward pass vs. N forward passes); our method produces multi-views with a single forward pass, whereas data augmentation typically requires N forward passes for N augmented inputs, leading to significantly higher computational cost, and 2) Evenly dispersed representations; by explicitly maximizing the angular diversity in the representation space, our method ensures that augmented views are evenly dispersed and exhibit minimal redundancy. In contrast, heuristic data augmentation influences embeddings only indirectly, often resulting in redundant representations or ones that drift away from the teacher's original view, making it difficult to achieve consistent and balanced dispersion in the representation space.

# E   Additional Theoretical Analysis

In this section, we supply the full derivations to support our theoretical claims in Sec. 3. Theoretical Analysis of the main manuscript. We begin by deriving the upper bound of the expected ensemble loss using the ensemble diversity measure [38] and then explicitly connect these to our inter- and intra-angle diversity objectives.

Following the analysis in [38], we first show that the expected loss $\mathbb{E}_{(\mathbf{x},\mathbf{y})\sim\mathcal{D}}[\mathcal{L}(\cdot)]$ of the ensemble logit $\mathbf{Z}^E$ (*i.e.*, $\mathbb{E}_{i=0}^N[\mathbf{Z}_i]$ for all ensemble members $\{\mathbf{Z}_i\}_{i=0}^N$) over the dataset $\mathcal{D}$ is bounded as follows:

$$\mathbb{E}_{(\mathbf{x},\mathbf{y})\sim\mathcal{D}}\left[\mathcal{L}(\mathbf{Z}^E)\right] \leq \mathbb{E}_{(\mathbf{x},\mathbf{y})\sim\mathcal{D}}\left[\mathbb{E}_{i=0}^N\left[\mathcal{L}(\mathbf{Z}_i)\right]\right] - K\mathbb{D}_{\text{Generalized}}(\{\mathbf{Z}_i\}_{i=0}^N), \quad (13)$$

where $\mathcal{L}$ is a KL divergence loss and $K > 0$ is a constant. $\mathbb{D}_{\text{Generalized}}$ is the same as the ensemble diversity $\mathbb{D}$ that appears in the main paper (the reason why we call this as generalized diversity are present in Sec. E.1). This bound implies that increasing the generalized ensemble diversity $\mathbb{D}$ $\mathbb{D}_{\text{Generalized}}$ lowers the upper bound of the expected ensemble loss. We also show that the generalized ensemble diversity can be expressed in terms of the total variance of logits across classes (see Sec. E.1 for the full formulation):

$$\mathbb{D}_{\text{Generalized}}(\{\mathbf{Z}_i\}_{i=0}^N) = \mathbb{E}_{(\mathbf{x},\mathbf{y})\sim\mathcal{D}}\left[\mathbb{V}_{i=0}^N\left[\frac{\mathbf{Z}_i}{\max \mathbf{Z}_i}\right]\right]. \quad (14)$$

Here, we define the total variance by $\mathbb{V}$ over all $C$ classes as $\mathbb{V}[\mathbf{z}] = \text{Var}_{\text{total}}[\mathbf{z}] = \sum_{j=1}^C \text{Var}[z^j]$ where $\mathbf{z} = [z^1, \cdots, z^C]$.

Furthermore, we show that this formulation is approximately proportional to the expected total variance of the logits (see Sec. E.2 for detailed derivation):

$$\mathbb{D}_{\text{Generalized}}(\{\mathbf{Z}_i\}_{i=0}^N) = \mathbb{E}_{(\mathbf{x},\mathbf{y})\sim\mathcal{D}}\left[\mathbb{V}_{i=0}^N\left[\frac{\mathbf{Z}_i}{\max \mathbf{Z}_i}\right]\right] \propto \mathbb{E}_{(\mathbf{x},\mathbf{y})\sim\mathcal{D}}\left[\mathbb{V}_{i=0}^N\left[\mathbf{Z}_i\right]\right]. \quad (15)$$

This implies that increasing expected total variance raises the generalized ensemble diversity, resulting in a decrease of the upper bound of the expected ensemble loss.

Based on the expected total variance of logits in Eq. (36), we derive two mathematically equivalent formulations of ensemble diversity in terms of angular similarity. Specifically, we show that the ensemble diversity measure $\mathbb{D}_{\text{Generalized}}(\cdot)$ can be expressed in the following two equivalent forms: 1) Inter-angle form (Eq. (16a)) uses the pairwise cosine similarity $s_{ij}^A$ (defined in Eq. (3) as inter-angle similarity) between each pair of augmented logits $\mathbf{Z}_i$; 2) Intra-angle form (Eq. (16b)) uses the cosine similarity $s_{ij}^{\triangle}$ (defined in Eq. (4) as intra-angle similarity) between the difference vectors $\Delta_i^{T-A} = \mathbf{Z}^T - \mathbf{Z}_i^A$, as follows:

$$
\mathbb{D}_{\text{Generalized}}(\{\mathbf{Z}_i\}_{i=0}^N) = \mathbb{E}_{(\mathbf{x},\mathbf{y})\sim\mathcal{D}}\left[\mathbb{V}_{i=0}^N\left[\frac{\mathbf{Z}_i}{\max \mathbf{Z}_i}\right]\right] =
\begin{cases}
\underbrace{\mathbb{E}_{(\mathbf{x},\mathbf{y})\sim D}\left[\mathbb{E}_{i=1}^N\left[\|\mathbf{Z}_i\|^2\right] - \mathbb{E}_{i,j=1}^N\left[\|\mathbf{Z}_i\|\|\mathbf{Z}_j\|s_{ij}^A\right]\right]}_{\text{Inter-angle Formulation}}, & (16a) \\[2em]
\underbrace{-\mathbb{E}_{(\mathbf{x},\mathbf{y})\sim D}\left[\mathbb{E}_{i,j=1}^N\left[\|\Delta_i^{T-A}\|\|\Delta_j^{T-A}\|s_{ij}^{\triangle}\right]\right]}_{\text{Intra-angle Formulation}}, & (16b)
\end{cases}
$$

where the detailed derivation is provided in Sec. E.3:

### E.1  Generalized Ensemble Diversity beyond a Single Class

The original ensemble diversity metric in [38] measures diversity only on logit $z_i^y$ for the ground-truth class $y$. We generalize this to include all $C$ logits $\mathbf{Z}_i = [z_i^1, \ldots, z_i^C]$, computing diversity across every class logit. This full-spectrum diversity measure appears in both our main paper and supplemental proofs. Compared to the original single-class version [38], the generalized metric introduces a constant scaling $K$ because it sums over $C$ classes instead of one target class. Below, we detail the derivation of this generalized ensemble diversity.

**Original Ensemble Diversity.**  The classic ensemble diversity [38] is defined as follows:

$$
\mathbb{D}_{\text{Original}}(\{\mathbf{Z}_i\}_{i=0}^N) = \mathbb{E}_{(\mathbf{x},\mathbf{y})\sim\mathcal{D}}\left[\text{Var}_i\left[\frac{z_i^y}{\max_k z_k^y}\right]\right], \quad \text{where } z_i^y := [\mathbf{Z}_i]^y, \tag{17}
$$

and $[\mathbf{Z}_i]^y$ is the logit of the target class $y$ in the vector $\mathbf{Z}_i$ among $C$ classes. This diversity focuses solely on the ground-truth class logit $z_i^y$.

**Generalized Ensemble Diversity.**  We extend this to capture variability across all $C$ classes as:

$$
\mathbb{D}_{\text{Generalized}}(\{\mathbf{Z}_i\}_{i=0}^N) = \mathbb{E}_{(\mathbf{x},\mathbf{y})\sim\mathcal{D}}\left[\mathbb{V}_{i=0}^N\left[\frac{\mathbf{Z}_i}{\max \mathbf{Z}_i}\right]\right], \tag{18}
$$

where $\max\mathbf{Z}_i$ denotes the maximal logit value over all class indices $j = 1, \ldots, C$. This formulation captures diversity across every output logit rather than focusing solely on the target class. This full-spectrum diversity metric underpins our theoretical analysis in both the main manuscript and the supplementary. The complete derivation follows below.

**Detailed derivation.**  The original diversity metric [38] is derived from the cross-entropy loss in their paper, so it only measures variability in the ground-truth class logit. In contrast, our generalized diversity is based on the KL-divergence loss between softmax distributions, capturing variability across all $C$ class logits rather than just the target class. For the derivation, we begin with the Taylor expansion of the logarithmic function $\log(x)$ around a fixed point $a > 0$, incorporating the second-order remainder term:

$$
\log x = \log a + \frac{1}{a}(x - a) - \frac{1}{2\xi^2}(x - a)^2, \quad \xi \in (x, a). \tag{19}
$$

Here, $\xi$ lies between $x$ and $a$. This expansion will allow us to approximate log terms arising in the KL-divergence expressions. Applying Eq. (19) with $x = \frac{z_i^j}{y^j}$ centering at $\mathbb{E}_{i=0}^N[\frac{z_i^j}{y^j}] = \frac{\mathbb{E}_{i=0}^N[z_i^j]}{y^j} > 0$, for an arbitrary ensemble member $i$, class index $j$, ground-truth label $\mathbf{Y} = [y^1, \ldots, y^C]$ with $y^j > 0$

and ensembling $\mathbb{E}_{i=0}^N[z_i^j]$ with uniform weights over $N$ ensemble units for specific $j$-th category logit, we have:

$$\log(\frac{z_i^j}{y^j}) = \log(\frac{\mathbb{E}_{i=0}^N[z_i^j]}{y^j}) + \frac{1}{\mathbb{E}_{i=0}^N[z_i^j]}(z_i^j - \mathbb{E}_{i=0}^N[z_i^j]) - \frac{1}{2\xi^2(y^j)^2}(z_i^j - \mathbb{E}_{i=0}^N[z_i^j])^2. \qquad (20)$$

By taking the expectation on both sides and multiplying through by $y^j$, we obtain:

$$\mathbb{E}_{i=0}^N[y^j \log(\frac{z_i^j}{y^j})] = y^j \log(\frac{\mathbb{E}_{i=0}^N[z_i^j]}{y^j}) - \mathbb{E}_{i=0}^N\big[\frac{1}{2\xi^2 z_{GT}^j}(z_i^j - \mathbb{E}_{i=0}^N[z_i^j])^2\big]. \qquad (21)$$

Rearranging terms,

$$y^j \log(\frac{y^j}{\mathbb{E}_{i=0}^N[z_i^j]}) = \mathbb{E}_{i=0}^N[y^j \log(\frac{y^j}{z_i^j})] - \mathbb{E}_{i=0}^N\big[\frac{1}{2\xi^2 y^j}(z_i^j - \mathbb{E}_{i=0}^N[z_i^j])^2\big]. \qquad (22)$$

Next, observe that since $\xi \in (x, a)$, it follows that $\xi \le \max x = \max_i \frac{z_i^j}{y^j}$. Using this bound, we derive the inequality:

$$y^j \log(\frac{y^j}{\mathbb{E}_{i=0}^N[z_i^j]}) \le \mathbb{E}_{i=0}^N[y^j \log(\frac{y^j}{z_i^j})] - \mathbb{E}_{i=0}^N\big[\frac{y^j}{2(\max_i z_i^j)^2}(z_i^j - \mathbb{E}_{i=0}^N[z_i^j])^2\big]. \qquad (23)$$

Summing over all classes $j \in \{1, \ldots, C\}$, we obtain:

$$\sum_{j=1}^C y^j \log(\frac{y^j}{\mathbb{E}_{i=0}^N[z_i^j]}) \le \mathbb{E}_{i=0}^N[\sum_{j=1}^C y^j \log(\frac{y^j}{z_i^j})] - \mathbb{E}_{i=0}^N\big[\sum_{j=1}^C \frac{y^j}{2(\max_i z_i^j)^2}(z_i^j - \mathbb{E}_{i=0}^N[z_i^j])^2\big]. \qquad (24)$$

Recalling the definition of KL-divergence $KL(\mathbf{Y}||\mathbf{Z}_i) = \sum_{j=1}^C y^j \log\frac{y^j}{z_i^j}$, the above inequality can be rewritten as:

$$KL(\mathbf{Y}||\mathbb{E}_{i=0}^N[\mathbf{Z}]) \le \mathbb{E}_{i=0}^N[KL(\mathbf{Y}||\mathbf{Z}_i)] - \mathbb{E}_{i=0}^N\big[\sum_{j=1}^C \frac{y^j}{2(\max_i z_i^j)^2}(z_i^j - \mathbb{E}_{i=0}^N[z_i^j])^2\big]. \qquad (25)$$

Taking expectation over data distribution $(\mathbf{x}, \mathbf{y}) \sim \mathcal{D}$ on both sides leads to:

$$\mathbb{E}_{(\mathbf{x},\mathbf{y})\sim\mathcal{D}}\Big[KL(\mathbf{Y}||\mathbb{E}_{i=0}^N[\mathbf{Z}])\Big]$$
$$\le \mathbb{E}_{(\mathbf{x},\mathbf{y})\sim\mathcal{D}}\mathbb{E}_{i=0}^N[KL(\mathbf{Y}||\mathbf{Z}_i)] - \mathbb{E}_{(\mathbf{x},\mathbf{y})\sim\mathcal{D}}\Big[\mathbb{E}_{i=0}^N\big[\sum_{j=1}^C \frac{y^j}{2(\max_i z_i^j)^2}(z_i^j - \mathbb{E}_{i=0}^N[z_i^j])^2\big]\Big]. \qquad (26)$$

We now recall the definition of the generalized diversity measure:

$$\mathbb{D}_{\text{Generalized}}(\{\mathbf{Z}_i\}_{i=0}^N) = \mathbb{E}_{(\mathbf{x},\mathbf{y})\sim\mathcal{D}}\Big[\mathbb{V}_{i=0}^N\big[\frac{\mathbf{Z}_i}{\max \mathbf{Z}_i}\big]\Big] = \mathbb{E}_{(\mathbf{x},\mathbf{y})\sim\mathcal{D}}\Big[\sum_{j=1}^C \text{Var}_i\big[\frac{z_i^j}{\max_k z_k^j}\big]\Big] \qquad (27)$$

$$= \mathbb{E}_{(\mathbf{x},\mathbf{y})\sim\mathcal{D}}\Big[\mathbb{E}_{i=0}^N\big[\sum_{j=1}^C \frac{1}{(\max_i z_i^j)^2}(z_i^j - \mathbb{E}_{i=0}^N[z_i^j])^2\big]\Big]. \qquad (28)$$

Since $y^j \ge 0$, the last term in Eq. (26) corresponds to the summation of the generalied diversity measure $\mathbb{D}_{\text{Generalized}}(\{\mathbf{Z}_i\}_{i=0}^N)$ in Eq. (27) over all class indices $\forall j \in \{1, \ldots, C\}$. (we insert this constant sum into a constant factor $K$, resulting in $\mathbb{D}_{\text{Original}}(\cdot) = K\mathbb{D}_{\text{Generalized}}(\cdot)$). Therefore, maximizing the diversity $\mathbb{D}_{\text{Generalized}}(\{\mathbf{Z}_i\}_{i=0}^N)$ increases the negative term on the right-hand side of Eq. (26), which tightens the upper bound and effectively reduces the expected KL-divergence loss of the ensemble.

**Relations between Expected Ensemble Loss and Generalized Diversity.** The original ensemble diversity work [38] shows that the expected ensemble loss of the ensemble logit $\mathbb{E}_{i=0}^N[\mathbf{Z}_i]$ is bounded by:

$$\mathbb{E}_{(\mathbf{x},\mathbf{y})\sim\mathcal{D}}\Big[\mathcal{L}(\mathbb{E}_{i=0}^N[\mathbf{Z}_i])\Big] \leq \mathbb{E}_{(\mathbf{x},\mathbf{y})\sim\mathcal{D}}\Big[\mathbb{E}_{i=0}^N\big[\mathcal{L}(\mathbf{Z}_i)\big]\Big] - \mathbb{D}_{\text{Original}}(\{\mathbf{Z}_i\}_{i=0}^N), \tag{29}$$

where $\mathcal{L}$ is a cross-entropy loss. Because our generalized diversity $\mathbb{D}_{\text{Generalized}}$ differs from the original one by a constant factor $K$, we can re-write this bound as:

$$\mathbb{E}_{(\mathbf{x},\mathbf{y})\sim\mathcal{D}}\Big[\mathcal{L}(\mathbb{E}_{i=0}^N[\mathbf{Z}_i])\Big] \leq \mathbb{E}_{(\mathbf{x},\mathbf{y})\sim\mathcal{D}}\Big[\mathbb{E}_{i=0}^N\big[\mathcal{L}(\mathbf{Z}_i)\big]\Big] - K\mathbb{D}_{\text{Generalized}}(\{\mathbf{Z}_i\}_{i=0}^N), \tag{30}$$

which is Eq. (6) in the main paper. It implies that increasing the ensemble diversity $\mathbb{D}$ tightens the upper bound on the ensemble's expected loss, thereby delivering a strong supervisory signal to the student and improving KD performance, as demonstrated in [34]. Empirically, Table 16 and Table 15 demonstrate that higher ensemble diversity indeed correlates with a lower measured expected ensemble loss, confirming this theoretical relationship.

### E.2 Ensemble Diversity is Proportional to Logit Variance

In Eq. (36), we demonstrate that the generalized ensemble diversity is a function of the total logit variance. Below, we offer the detailed derivation, showing that the generalized ensemble diversity is a scaled total logit variance. We start from the definition of the generalized diversity measure:

$$\mathbb{D}_{\text{Generalized}}(\{\mathbf{Z}_i\}_{i=0}^N) = \mathbb{E}_{(\mathbf{x},\mathbf{y})\sim\mathcal{D}}\Big[\mathbb{V}_{i=0}^N\big[\frac{\mathbf{Z}_i}{\max \mathbf{Z}_i}\big]\Big] \tag{31}$$

$$= \mathbb{E}_{(\mathbf{x},\mathbf{y})\sim\mathcal{D}}\Big[\sum_{j=1}^C \text{Var}_i\big[\frac{z_i^j}{\max_k z_k^j}\big]\Big] = \mathbb{E}_{(\mathbf{x},\mathbf{y})\sim\mathcal{D}}\Big[\sum_{j=1}^C \frac{\text{Var}_i[z_i^j]}{(\max_k z_k^j)^2}\Big]. \tag{32}$$

To analyze the relationship between diversity and total logit variance, we model $z_i^j$ across all ensemble units $\{i\}_{i=0}^N$ as a Gaussian variable with mean $\mu_j = \mathbb{E}_{i=0}^N[z_i^j]$ and variance $\sigma_j^2$. Since $z_i^j$ represents a logit probability for a specific category $j$, it is bounded above by 1. Based on this assumption, we approximate the maximum value over individual unit $i$ as:

$$\max_i z_i^j \approx \min(\mathbb{E}_{i=0}^N[z_i^j] + k_j\sigma_j, 1) \tag{33}$$

Which approximates the maximum values of $z_i^j$ across $i$ by capturing deviations above the mean under the Gaussian assumption, while the minimum operator ensures the value remains bounded by 1. Thus, we can rewrite the generalized diversity measure as:

$$\mathbb{D}(\{\mathbf{Z}_i\}_{i=0}^N) = \mathbb{E}_{(\mathbf{x},\mathbf{y})\sim\mathcal{D}}\Big[\sum_{j=1}^C \frac{\text{Var}_i[z_i^j]}{(\max_k z_k^j)^2}\Big] = \mathbb{E}_{(\mathbf{x},\mathbf{y})\sim\mathcal{D}}\Big[\sum_{j=1}^C \frac{\sigma_j^2}{\min(\mathbb{E}_{i=0}^N[z_i^j] + k_j\sigma_j, 1)^2}\Big] \tag{34}$$

To show that diversity increases monotonically with variance, it suffices to prove that the following function is monotonically increasing in $\sigma$.

$$f(\delta) = \frac{\sigma^2}{\min(p + k\sigma, 1)^2} \quad, \text{ where } p \in [0,1], k \geq 0. \tag{35}$$

We consider two cases:

**Case 1:** If $\min(p + k\sigma, 1) = 1$, then $f(\sigma) = \sigma^2$, which is clearly increasing in $\sigma$.

**Case 2:** If $\min(p + k\sigma, 1) = p + k\sigma$, then:

$$f(\sigma) = \frac{\sigma^2}{(p + k\sigma)^2}, \quad f'(\sigma) = \frac{2p\sigma}{(p + k\sigma)^3} \geq 0$$

Since $p \in [0,1], \sigma \geq 0$ and the denominator is always non-negative, $(p + k\sigma)^3 \geq 0$, we have $f'(\sigma) \geq 0$. Therefore, in both cases, $f(\sigma)$ is a non-decreasing function of $\sigma$. Thus, increasing the variance $\sigma_j^2$ results in an increased diversity measure $\mathbb{D}(\{\mathbf{Z}_i\}_{i=0}^N)$, thereby supporting our claim that ensemble diversity is proportional to logit variance.

### E.3 Two Equivalent Forms of Ensemble Diversity

In this section, we show that the generalized ensemble diversity metric $\mathbb{D}_{\text{Generalized}}(\cdot)$ can be written in two mathematically equivalent forms: 1) Inter-angle form (Eq. (16a)) and 2) Intra-angle form (Eq. (16b)). We start from the total variance of all ensemble logits $\{\mathbf{Z}_i\}_{i=0}^N$ for two form derivations, since the ensemble diversity is proportional to the total logit variance, as follows and proved in Sec E.2:

$$\mathbb{D}_{\text{Generalized}}(\{\mathbf{Z}_i\}_{i=0}^N) = \mathbb{E}_{(\mathbf{x},\mathbf{y})\sim\mathcal{D}}\Big[\mathbb{V}_{i=0}^N\big[\frac{\mathbf{Z}_i}{\max \mathbf{Z}_i}\big]\Big] \propto \mathbb{E}_{(\mathbf{x},\mathbf{y})\sim\mathcal{D}}\Big[\underbrace{\mathbb{V}_{i=0}^N\big[\mathbf{Z}_i\big]}_{\text{Start of derivation}}\Big]. \tag{36}$$

**Inter-angle Formulation of Ensemble Diversity.** To derive the inter-angle form (Eq. (16a)) of diversity from the total logit variance, *i.e.*, $\mathbb{V}_{i=0}^N[\mathbf{Z}_i]$, we focus solely on the augmented logits $\{\mathbf{Z}_i\}_{i=1}^N$, omitting the teacher ($i=0$) since our angular objectives apply to these views. We then begin with a definition of variance to derive the inter-angle form (Eq. (16a)) from the logit variance, *i.e.*, $\mathbb{V}_{i=1}^N[\mathbf{Z}_i]$:

$$\mathbb{V}_{i=1}^N\big[\mathbf{Z}_i\big] = \mathbb{E}_{i=1}^N\big[\|\mathbf{Z}_i\|^2\big] - \|\mathbb{E}_{i=1}^N\big[\mathbf{Z}_i\big]\|^2 \tag{37}$$

$$= \frac{1}{N}\sum_{i=1}^N \|\mathbf{Z}_i\|^2 - \|\frac{1}{N}\sum_{i=1}^N \mathbf{Z}_i\|^2 \tag{38}$$

$$= \frac{1}{N}\sum_{i=1}^N \|\mathbf{Z}_i\|^2 - \frac{1}{N^2}\sum_{i=1}^N\sum_{j=1}^N \mathbf{Z}_i \cdot \mathbf{Z}_j \tag{39}$$

$$= \frac{1}{N}\sum_{i=1}^N \|\mathbf{Z}_i\|^2 - \frac{1}{N^2}\sum_{i=1}^N\sum_{j=1}^N \|\mathbf{Z}_i\|\|\mathbf{Z}_j\|\cos(\mathbf{Z}_i,\mathbf{Z}_j) \tag{40}$$

$$= \frac{1}{N}\sum_{i=1}^N \|\mathbf{Z}_i\|^2 - \frac{1}{N^2}\sum_{i=1}^N\sum_{j=1}^N \|\mathbf{Z}_i\|\|\mathbf{Z}_j\|s_{ij}^A \tag{41}$$

$$= \underbrace{\mathbb{E}_{i=1}^N\big[\|\mathbf{Z}_i\|^2\big] - \mathbb{E}_{i,j=1}^N\big[\|\mathbf{Z}_i\|\|\mathbf{Z}_j\|s_{ij}^A\big]}_{\text{Inter-angle Formulation in Eq (16a)}}, \tag{42}$$

where $s_{ij}^A = \cos(\mathbf{Z}_i,\mathbf{Z}_j)$, $\cos(\mathbf{u},\mathbf{v}) = \mathbf{u}\cdot\mathbf{v}/\|\mathbf{u}\|\|\mathbf{v}\|$, and $\mathbb{V}$ denotes a total variance over all $C$ classes. The inter-angle formulation of the ensemble diversity makes it clear that reducing each pairwise similarity $s_{ij}^A$ increases the total variance of logits and therefore raises the ensemble diversity. Our constrained inter-angle diversity loss explicitly minimizes $s_{ij}^A$, so it naturally enhances the ensemble diversity $\mathbb{D}$. In Table 15, we empirically observe that adding the inter-angle diversity loss both raises the average inter-angle (*i.e.*, lowers $s_{ij}^A$) and increases the ensemble diversity measure.

**Intra-angle Formulation of Ensemble Diversity.** We can express $\mathbb{V}_{i=1}^N[\mathbf{Z}_i]$ depending on $s_{ij}^\Delta$, using the property that $\sum_{i=1}^N \Delta_i^{T-A} = \mathbf{0}$ (we assume that the average of augmented logits can approximate the original teacher $\mathbf{Z}^T = \frac{1}{N}\sum_{i=1}^N \mathbf{Z}i$). We also start from the definition of variance for the derivation of inter-angle formulation (Eq. (16b)):

$$\mathbb{V}_{i=1}^N[\mathbf{Z}_i] = \mathbb{E}_{i=1}^N\big[\|\mathbf{Z}_i\|^2\big] - \|\mathbb{E}_{i=1}^N[\mathbf{Z}_i]\|^2 \tag{43}$$

$$= \frac{1}{N}\sum_{i=1}^N\|\mathbf{Z}_i\|^2 - \|\frac{1}{N}\sum_{i=1}^N\mathbf{Z}_i\|^2 \tag{44}$$

$$= \frac{1}{N}\sum_{i=1}^N\|\mathbf{Z}_i\|^2 - \frac{1}{N^2}\sum_{i=1}^N\sum_{j=1}^N\mathbf{Z}_i\cdot\mathbf{Z}_j \tag{45}$$

$$= \frac{1}{N}\sum_{i=1}^N\|\mathbf{Z}_i\|^2 - \frac{2}{N^2}\sum_{i=1}^N\sum_{j=1}^N\mathbf{Z}_{\mathbf{i}}\cdot\mathbf{Z}_j + \frac{1}{N^2}\sum_{i=1}^N\sum_{j=1}^N\mathbf{Z}_{\mathbf{i}}\cdot\mathbf{Z}_j \tag{46}$$

$$= \frac{1}{N}\sum_{i=1}^N\|\mathbf{Z}_i - \mathbf{Z}^T\|^2 \qquad (\because \mathbf{Z}^T = \frac{1}{N}\sum_{i=1}^N\mathbf{Z}_i) \tag{47}$$

$$= \frac{1}{N}\sum_{i=1}^N\|\mathbf{\Delta}_i^{T-A}\|^2 \tag{48}$$

$$= \frac{1}{N}[(\sum_{i=1}^M\mathbf{\Delta}_i^{T-A})\cdot(\sum_{i=1}^N\mathbf{\Delta}_i^{T-A}) - \sum_{i=1}^N\sum_{j=1}^N\mathbf{\Delta}_i^{T-A}\cdot\mathbf{\Delta}_j^{T-A}] \tag{49}$$

$$= \frac{1}{N}[\mathbf{0}\cdot\mathbf{0} - \sum_{i=1}^N\sum_{j=1}^N\mathbf{\Delta}_i^{T-A}\cdot\mathbf{\Delta}_j^{T-A}] \tag{50}$$

$$= -\frac{1}{N}\sum_{i=1}^N\sum_{j=1}^N\mathbf{\Delta}_i^{T-A}\cdot\mathbf{\Delta}_j^{T-A} \tag{51}$$

$$= -\frac{1}{N}\sum_{i=1}^N\sum_{j=1}^N\|\mathbf{\Delta}_i^{T-A}\|\|\mathbf{\Delta}_j^{T-A}\|s_{ij}^{\mathbf{\Delta}}. \tag{52}$$

$$= \underbrace{-\mathbb{E}_{i,j=1}^N\big[\|\mathbf{\Delta}_i^{T-A}\|\|\mathbf{\Delta}_j^{T-A}\|s_{ij}^{\mathbf{\Delta}}\big]}_{\text{Intra-angle Formulation in Eq (16b)}}. \tag{53}$$

Since lowering each $s_{ij}^{\mathbf{\Delta}}$ raises the variance of the logits, our intra-angle diversity loss, which minimizes $s_{ij}^{\mathbf{\Delta}}$, directly increases both variance and, consequently, the ensemble diversity. Table 15 empirically confirms this: adding intra-angle diversity loss not only boosts the mean intra-angle between views but also elevates the measured ensemble diversity.

We validate our two core theoretical claims: 1) our angular-diversity objectives boost ensemble diversity, as derived in Sec. E.3, and 2) increased diversity tightens the ensemble loss bound, as proven in Sec. E.1. Empirically, Table 15 shows that our inter- and intra-angle diversity losses both raise the measured ensemble diversity and lowers the measured ensemble expected loss, directly translating into stronger KD performance.

### E.4  Better Teacher, Better Student

In this section, we leverage prior theoretical results [34, 41, 9, 16] to connect improvements in the teacher's expected loss (in our case, the ensemble's expected loss) directly to the reductions in the student's expected loss. Specifically, in [34], they derive the student's generalization error (measured as the discrepancy between its KD loss ($\mathcal{L}_{\text{KD}}$) and the KL loss ($\mathcal{L}_{\text{KL}}$) to a Bayes-optimal soft label $\mathbf{Y}$ can be bounded by two terms: 1) the variance of the KD loss across the data, and 2) the teacher's approximation error to the ideal soft label (captured by an MSE term). Formally,

$$\underbrace{\mathbb{E}_{(\mathbf{x},\mathbf{y})\sim\mathcal{D}}\Big[\big(\mathcal{L}_{\text{KD}}(\mathbf{Z}^T,\mathbf{Z}^S) - \mathcal{L}_{\text{KL}}(\mathbf{Y},\mathbf{Z}^S)\big)^2\Big]}_{\text{Student Generalization Error (KD loss - GT loss)}} \leq \frac{1}{N}\underbrace{\mathbb{V}_{(\mathbf{x},\mathbf{y})\sim\mathcal{D}}\big[\mathcal{L}_{\text{KD}}(\mathbf{Z}^T,\mathbf{Z}^S)\big]}_{\text{Variance of Distillation Loss}} + \kappa\cdot\underbrace{\mathbb{E}_{(\mathbf{x},\mathbf{y})\sim\mathcal{D}}\big[\mathcal{L}_{\text{MSE}}(\mathbf{Z}^T,\mathbf{Y})\big]^2}_{\text{Teacher Approximate Loss}},$$

where $\mathbf{Z}^T$ and $\mathbf{Z}^S$ are the output logits of the teacher and student, respectively, $\mathbf{Y}$ is a soft ideal ground-truth label (the Bayes-optimal target distribution), and $\kappa$ is a positive constant value.

Employing knowledge distillation could reduce the first term by providing the student with a consistent teacher signal, which lowers the variability of the KD loss across the dataset $\mathcal{D}$. The second term is decreased by using an ensemble: generating diverse views (*i.e.*, ensemble) produces a closer approximation to the ideal soft target (the Bayes-optimal target distribution), reducing the teacher's approximation error. Together, these effects tighten the upper bound on the student's generalization error, confirming that stronger teacher signals (via our angular-KD augmentations) directly improve the student's performance.

