# OpenReview forum: "Single-Teacher View Augmentation: Boosting Knowledge Distillation via Angular Diversity"
_NeurIPS.cc/2025/Conference — NeurIPS 2025 poster_

### Official Review · Reviewer_KCJt · 2025-06-05

**Clarity:** 3
**Significance:** 2
**Originality:** 3
**Rating:** 5
**Confidence:** 4

**Summary:**

This paper presents a new knowledge distillation method.  The idea is motivated from recent work showing that using multiple teacher networks for training a single student network leads to improved accuracy.  However, using multiple teacher networks is computationally expensive.  The new idea is to add "view augmentation heads" to a single teacher network to yield a similar effect to having multiple teacher networks with only a modest increase in computational cost.  View augmentation heads are linear transformations of either the final feature layer or the logit layer of the teacher network.  These new linear transformation layers are trained to produce new features or logits that exhibit diversity by using two new losses: an inter- and an intra-angle diversity loss.  The effect is similar to having multiple independent teacher networks since the augmentation heads yield diverse features and logits that are not too far from the original teacher's features and logits.  Experiments are conducted on the CIFAR-100, STL-10 and TinyImageNet datasets using various teacher-student network pairs.  Modest improvements are shown over a competing view augmentation approach.

**Questions:**

More discussion of how this paper's improvement to knowledge distillation compares with the many other papers on knowledge distillation improvements, not just multiple teacher approaches, would be helpful.

Why weren't all the experiments in Table 1 averaged over 3 or more runs?

**Ethical Concerns:**

["NO or VERY MINOR ethics concerns only"]

**Final Justification:**

The authors did a good job of addressing my concerns by providing standard deviations for results over multiple runs and by including more comparisons to other KD methods.  I think the paper is above the bar for acceptance to NeurIPS.

**Limitations:**

Yes

**Paper Formatting Concerns:**

I don't have any paper formatting concerns.

**Quality:**

3

**Strengths And Weaknesses:**

The main idea of the paper, simulating multiple teacher networks using low-cost view augmentation heads, is well motivated and makes sense.

The paper is written clearly and is relatively easy to follow.

The proposed method is novel and includes new loss functions and a novel approach to view augmentation for improving knowledge distillation.

The theoretical results showing that increased ensemble diversity lowers the upper bound on the ensemble loss, is helpful for understanding how the view augmentation heads improve accuracy.

Why don't the experiments, especially in Table 1 average over multiple runs and include standard deviations?  This has been done in many past knowledge distillation papers.  The appendix A.4 does show one experiment that is averaged over 3 runs, but it would be even better to do this on all or most of the experiments.

The paper does not give a good sense of how the proposed view augmentation approach fits in with the large amount of previous work on knowledge distillation in terms of accuracy and computational cost.  There have been many proposed modifications to the original knowledge distillation scheme of Hinton et al. 2015, each of which typically improves accuracy a modest amount.  Since this paper only compares with TeKAP, it is unclear how it compares to the hundreds (thousands?) of other modifications to knowledge distillation that have been proposed.

Minor typo:  On line 314, "Hiton" should be "Hinton".

---

> ### Author Rebuttal · Authors · 2025-07-29
>
> Dear reviewer KCJt,
>
> Thank you for your time and efforts in reviewing our manuscript. We greatly appreciate your recognition of the strengths of our work, including the well-motivated and novel methodology, the easy-to-follow presentation, and the theoretical results that help explain how our approach improves accuracy.
>
> ---
>
> **[W1, Q2] Request for results averaged over three runs.**
>
> We fully agree that averaging over multiple random seeds provides a more reliable evaluation. Our initial submission did not include three-run average results in Table 1 due to computational constraints (Table 1 spans 18 settings across architectures, and our experiments were conducted on limited resources of 2080 Ti GPUs). In the table below, we additionally report three-run averaged results under two representative configurations from Table 1:
> (1) a ResNet-32x4 teacher -> a ResNet-8x4 student, and
> (2) a ResNet32x4 teacher -> a WideResNet-40-2 student.
> These are presented alongside results from TeKAP, a SoTA knowledge augmentation method.
>
>
> $$
> \\begin{array}{lcc}
> \\textbf{Teacher} & \\text{ResNet-32}\\times\\text{4} & \\text{ResNet-32}\\times\\text{4} \\\\
> \\textbf{Student} & \\text{ResNet-8}\\times\\text{4} & \\text{WideResNet-40-2} \\\\ \\hline
> \\textcolor{gray}{\\text{Logit distil. (KD) w/o aug}} & \\textcolor{gray}{\\text{73.33}} & \\textcolor{gray}{\\text{77.70}} \\\\
> \\text{w/ TeKAP} & 75.84\\,\\pm 0.156 & 77.99\\,\\pm 0.087     \\\\
> \\text{w/ Angular-KD (ours)} & \\mathbf{76.08}\\,\\pm 0.217 & \\mathbf{78.57}\\,\\pm 0.125  \\\\ \hline
> \\textcolor{gray}{\\text{Feature distil. (CRD) w/o aug}} & \\textcolor{gray}{\\text{75.51}} & \\textcolor{gray}{\\text{78.15}} \\\\
> \\text{w/ TeKAP} & 75.82\\,\\pm 0.153& 77.97\\,\\pm 0.207     \\\\
> \\text{w/ Angular-KD (ours)} & \\mathbf{75.94}\\,\\pm 0.105 & \\mathbf{78.56}\\,\\pm 0.195  \\\\ \hline
> \\textcolor{gray}{\\text{Logit +  Feat. distil. (KD + CRD) w/o aug}} & \\textcolor{gray}{\\text{75.46}} & \\textcolor{gray}{\\text{78.01}} \\\\
> \\text{w/ TeKAP} & 75.89\\,\\pm 0.061& 77.96\\,\\pm 0.299     \\\\
> \\text{w/ Angular-KD (ours)} & \\mathbf{76.39}\\,\\pm 0.168 & \\mathbf{78.62}\\,\\pm 0.050  \\\\ \hline
> \\end{array}
> $$
>
> We will include the full results of Table 1 across all settings with three-run averages in the final version, if accepted.
>
> ---
>
> **[W2, Q2] More comparisons with other KD methods beyond TeKAP.**
>
> Thank you for the reviewer’s valuable comments. We understand that limiting the comparison to only TeKAP may not sufficiently demonstrate the general effectiveness of our approach across diverse knowledge distillation (KD) paradigms.
>
> To address this concern, we emphasize that our method is **framework-agnostic** and designed to be **plug-and-play**, making it compatible with a wide range of KD frameworks. In our experiments, we applied our method on top of several representative and diverse KD frameworks, including DKD [57], MLKD [20] and TAKD [33] in addition to conventional KD [17] and CRD [45], and observed consistent improvements over their baseline performances (see Table 2 and Table 11).
>
> Additionally, the table below shows further plug-and-play results from applying our methods to Review-KD [6], in comparison with TeKAP. Once again, our method achieves consistent improvements, and this suggests that our approach can complement, rather than replace, existing KD methods.
>
> $$
> \\begin{array}{lcc}
> \\textbf{Teacher} & \\text{ResNet-32}\\times\\text{4} & \\text{WideResNet-40-2} & \\text{ResNet-32}\\times\\text{4}    \\\\
> \\textbf{Student} & \\text{ResNet-8}\\times\\text{4} & \\text{WideResNet-40-1}  & \\text{WideResNet-16-2} \\\\ \\hline
> \\textcolor{gray}{\\text{Review-KD w/o aug}} & \\textcolor{gray}{\\text{75.63}} & \\textcolor{gray}{\\text{75.09}} & \\textcolor{gray}{\\text{76.11}} \\\\
> \\text{w/ TeKAP} & 75.32 &  75.15  &76.36     \\\\
> \\text{w/ Angular-KD (ours)} & \\mathbf{75.78} & \\mathbf{75.45} & \\mathbf{76.40}  \\\\
> \\end{array}
> $$
>
> We will revise the manuscript to clarify this point and better highlight the broad compatibility and effectiveness of our approach across various KD frameworks.
>
> ---
>
> **[W3] Typo on line 314.**
>
> Thank you for pointing out the typo. We will correct “Hiton” to “Hinton” on line 314 in the final version.

---

> > ### Comment · Reviewer_KCJt · 2025-08-04
> >
> > After reading the authors' rebuttal as well as the other reviews, I would like to keep my original "Accept" rating.  The authors did a nice job of responding to the reviewers' critiques.

---

> ### Author Response · Authors · 2025-08-08
>
> Dear reviewer KCJt,
>
> We truly appreciate your recognition of our efforts in addressing the reviewers’ critiques, and we’re glad to hear that our responses were clear and satisfactory.
>
> In the final version, we will make sure to present the relevant results and discussions more explicitly.
>
> Please don’t hesitate to reach out if you have any further questions or suggestions.
>
> Thank you very much,
>
> Authors.

---

### Official Review · Reviewer_SW6m · 2025-06-27

**Clarity:** 3
**Significance:** 3
**Originality:** 3
**Rating:** 4
**Confidence:** 5

**Summary:**

This paper proposes Angular-KD, a novel knowledge distillation approach that enhances a single teacher model with lightweight linear branches to generate multiple diverse views. These views are optimized using two angular diversity objectives—constrained inter-angle and intra-angle losses—to increase semantic variance while preserving alignment with the original teacher. The approach is theoretically motivated and empirically validated across several benchmarks, showing improvements over prior work such as TeKAP.

**Questions:**

See Weaknesses.

**Ethical Concerns:**

["NO or VERY MINOR ethics concerns only"]

**Final Justification:**

Thank you for your constructive and insightful comments on my manuscript. I have carefully considered your feedback, and I am pleased to inform you that your responses have addressed the majority of my concerns. Your suggestions have significantly clarified several aspects of the paper, and I now feel more confident in its contribution to the field.

As a result, I have revised my assessment and have decided to increase the score accordingly.

**Limitations:**

The experiments are limited to image classification. Can it works on other tasks?

**Paper Formatting Concerns:**

None.

**Quality:**

3

**Strengths And Weaknesses:**

Strengths
1. The idea of generating diverse teacher views using angular diversity is novel and provides a unique perspective on knowledge augmentation.
2. The proposed method can be easily integrated into existing KD frameworks.
3. The paper is well-structured and logically coherent.

Weaknesses
1. The intra-angle diversify loss encourages an even distribution of views around the original output. However, the similarity between different views and the original image may vary, making this assumption unreasonable.
2. The theoretical analysis cannot match the algorithm well. The main result—Upper Bound on Expected Loss of Ensemble in Equation (6)—has significant issues. The first term on the right-hand side of the bound increases with the number of augmented views N, implying that more views can lead to worse performance. However, there is no proof provided to show that the bound is tighter as N increases.
3. Adding multiple branches to the teacher model increases much computational cost. However, the paper does not provide a comparison of the training time between the teacher and student models.
4. All the coefficients of three loss are 1 and is it reasonable. Can you conduct an ablation for the coefficients ?
5. The experiments are limited to image classification. It is necessary to include additional experiments on image segmentation as well as results on the ImageNet-1K dataset.

---

> ### Author Rebuttal · Authors · 2025-07-27
>
> Dear reviewer SW6m,
>
> Thank you for your time and efforts in reviewing our manuscript. We appreciate your recognition of our paper’s strengths, particularly our novel approach using angular diversity, the compatibility of our methods, and a well-structured paper. Below, we would like to provide clarifications and address the concerns and limitations you raised.
>
> ---
>
> **[W1] The assumption that intra-angle diversity loss promotes an even spread of views is unreasonable.**
>
> We acknowledge the concern that varying similarities between different views and the teacher output could challenge the assumption of an even spread.
>  However, in our empirical analysis (Table 15 in the supplementary), we observe that the intra-angle diversity loss significantly increases pairwise intra-angles between views.
>  We interpret larger intra-angles as indicating that the views are more evenly distributed around the teacher representation, i.e., they are less clustered in a single direction, as also illustrated in Figure 1b. We believe this supports our claim that the intra-angle diversity loss encourages directional diversity among the views.
>
> Furthermore, we clarify that in this work, the term “even spread” is used in the sense of promoting representational diversity and reducing directional redundancy among views, rather than enforcing equal similarity to the teacher output. We will revise the final version to make this interpretation more explicit and avoid potential confusion.
>
>
> ---
>
>
> **[W2] The first term on the right-hand side of the bound increases with the number of augmented views N.**
>
> We would like to clarify that the first term on the right-hand side of the expected ensemble loss represents the mean of the losses across all augmented views, not their sum. Therefore, this term does not increase with the number of views N; rather, it reflects the average quality of individual views (e.g., each view’s cross-entropy loss with respect to the ground-truth label).
>
> We will revise the explanation around Eq. (6) in the final version to clearly distinguish this averaging behavior and avoid any potential ambiguity.
>
>
> ---
>
>
> **[W3] The training time comparisons.**
>
> Thank you for your great suggestions. We agree that including a comparison of training time would be valuable. The following table compares the training time of our methods with 1) naive KD + CRD without augmentation, 2) the augmented one by TeKAP (a SoTA knowledge augmentation method), and 3) multi-teacher methods (e.g., Ensemble distillation [3]), including computational demands (FLOPs and Params; reported in Table 5). This experiment is conducted with a WideResNet40-2 teacher and a WideResNet-16-2 student on a 2080 Ti GPU over 50,000 training images and a total of 240 epochs.
>
> | Method          | Training time (1 epoch) | Params (M) | FLOPs (M) |
> | --------------- |:------------:| :------------:| :------------:|
> |  w/o aug |     27s     | 2.26 | 329.02 |
> | TeKAP      |   32s    |  2.26  | 329.02 |
> | Ours           |    35s    |  2.40 | 329.19 |
> | Ensemble distillation | 47s |  11.28 | 1645.10 |
>
> We observe that our multiple branches incur only a marginal increase in both training time and computation cost (FLOPs and Params), since the branches are composed of lightweight linear heads. In the revision, we will include these results to clearly present the lightweight nature of our approach.
>
> ---
>
> **[W4] Ablations on the loss coefficients.**
>
> Yes, we appreciate your valuable suggestion. The table below reports an ablation study over the three loss coefficients.
>
> | Coefficients (Inter:Intra:GT)          | Top-1 Accuracy |
> | --------------- |:------------:|
> | 0.5:1:1 |    76.03      |
> |  1:0.5:1     |   75.99    |
> |  1:1:0.5     |    75.91   |
> |  0.5:0.5:1     |   75.74    |
> | 1:1:1 (reported)         |     **76.46**   |
>
> We observe that setting all coefficients to 1 yields the best performance.
> In contrast, reducing the coefficients for both inter‑ and intra‑angle diversity losses to 0.5 results in a substantial decrease in performance, suggesting that these two angular losses are critical.
> We will include this result in the final version to highlight the effect of each loss component.
>
> ---
>
> **[W5, L1] Experiments on other tasks and ImageNet-1k.**
>
> We appreciate the reviewer’s interest in evaluating broader applicability. However, we would like to clarify that results on ImageNet-1K are already included in our main manuscript—specifically in Table 10 of the supplementary material, where our approach outperforms the naive KD method without augmentation and a SoTA knowledge augmentation method, TeKAP.
>
> To further validate that our approach is effective in other tasks, we are currently conducting object detection or image segmentation experiments and exploring training recipes optimized for our environments. Given the time required for thorough evaluation, we kindly ask whether it would be acceptable to share the results during the reviewer–author discussion phase. We will update the results as soon as the experiments are complete.

---

> ### Author Response · Authors · 2025-08-05
>
> Dear Reviewer SW6m,
>
> Thank you for your patience and understanding while we completed the additional experiments on other tasks. To provide quantitative results within our computational constraints using NVIDIA 2080 Ti GPUs during the reviewer–author discussion phase, we conducted a binary segmentation task on the Carvana Image Masking dataset (5,088 training images and 100,064 test images) using a teacher U-Net (depth 32) and a student U-Net (depth 16) in the table below.
>
> | Method   | Dice  Loss  ↓   | IoU       |
> |----------|-----------|-----------|
> | KD       | 0.0218 | 94.8 |
> | + Ours   | **0.0208**  | **95.7** |
>
> Our approach shows clear improvements across all metrics over the naive knowledge distillation.
> In the final version, we will extend these results to COCO-stuff for the semantic segmentation or MS-COCO for the object detection, and incorporate implementation details for each task.
>
> ---
>
> If you have any further questions or suggestions, please do not hesitate to let us know.
>
> Thank you again for your time and feedback.
>
> Sincerely,
>
> Authors.

---

> ### Author Response · Authors · 2025-08-07
> **A gentle reminder for Reviewer SW6m**
>
> Dear Reviewer SW6m,
>
> We appreciate your constructive comments for helping us to improve our paper in many aspects.
>
> This is a gentle reminder since the reviewer-author discussion window is coming to a close.
>
> We would like to ask if you have any further questions regarding our submission paper so that we can still respond.
>
> Thanks.
>
> Authors.

---

> > ### Comment · Reviewer_SW6m · 2025-08-09
> >
> > I have revised my assessment and have decided to increase the score accordingly. Thank you for your constructive and insightful comments on my manuscript. I have carefully considered your feedback, and I am pleased to inform you that your responses have addressed the majority of my concerns. Your suggestions have significantly clarified several aspects of the paper, and I now feel more confident in its contribution to the field.

---

> > > ### Author Response · Authors · 2025-08-09
> > >
> > > Dear Reviewer SW6m,
> > >
> > > Thank you for your considerate re-evaluation.
> > >
> > > We sincerely appreciate your constructive and insightful comments.
> > >
> > > We will carefully incorporate your feedbacks to further improve the clarity and overall quality of the manuscript.
> > >
> > > Thank you very much,
> > >
> > > Authors.

---

### Official Review · Reviewer_TLMU · 2025-07-02

**Clarity:** 3
**Significance:** 2
**Originality:** 3
**Rating:** 4
**Confidence:** 4

**Summary:**

This paper proposes the Angular-KD framework, which generates diverse multi-view knowledge by attaching multiple lightweight linear branches to a single teacher model. It introduces a constrained inter-angle diversity loss and an intra-angle diversity loss. Experimental results show that Angular-KD outperforms existing knowledge augmentation methods across multiple knowledge distillation benchmarks and can be integrated as a plug-and-play module with other KD frameworks.

**Questions:**

Although the proposed method generates diverse views by attaching multiple branches, it still relies on a single teacher model. This means that if the teacher model itself has biases or limitations, these biases may be passed on to the student model. In contrast, multi-teacher methods can provide a broader perspective through multiple independently trained teacher models. In the comparative experiments, the paper compares the performance with multi-teacher distillation methods, showing that this method outperforms them. This result raises the question of whether this method remains leading when the complexity of both the teacher and student models increases. The paper lacks test results on larger-scale models.

**Ethical Concerns:**

["NO or VERY MINOR ethics concerns only"]

**Final Justification:**

The author responses have addressed most of my concerns. However, the concept of enforcing angular diversity for optimization has already been explored in other works. Therefore, I have decided to retain my score as borderline accept.

**Limitations:**

Yes. The authors acknowledge that this method relies on the teacher's representation and thus cannot introduce truly novel semantics beyond the teacher's knowledge. While it outperforms multi-teacher methods, it lacks the ability to capture independent and complementary signals from multiple teachers. Relying on a single teacher may propagate teacher biases, which could have potential implications in privacy-sensitive or surveillance-related applications.

**Quality:**

3

**Strengths And Weaknesses:**

Streangth:

1.	The method presents a novel angle for KD augment and is compatible with other knowledge distillation frameworks, designed as a plug-and-play solution.

2.	Experiments were conducted on multiple datasets and in imbalanced scenarios, achieving good performance.

3.	The paper provides theoretical analysis and implementation code, enhancing the credibility and practicality of the research.

Weakness:

1.	The comparison is limited to only one data augmentation method, lacking comparisons with other advanced methods.

2.	The comparison with multi-teacher methods is limited and not up-to-date, such as TAKD published in 2023, which may not be the latest multi-teacher method.

3.	The visualization analysis in this paper is relatively limited, only providing t-SNE plots to show the distribution of the teacher and augmented views. It fails to further demonstrate the model's performance across different perspectives through other visualization methods (e.g., feature space distribution or decision boundaries).

---

> ### Author Rebuttal · Authors · 2025-07-29
>
> Dear reviewer TLMU,
>
> Thank you for your time and efforts in reviewing our manuscript. We appreciate your recognition of our paper’s strengths, particularly our novel and compatible methods, comprehensive experiments on multiple datasets, and our theoretical analysis and codes. Below, we would like to provide clarifications and address the concerns and questions you raised.
>
> ---
>
> **[W1] The comparison is limited to only one data augmentation method, lacking comparisons with other advanced methods.**
>
> Thank you for the reviewer’s valuable comments. We understand that limiting the comparison to only TeKAP may not sufficiently demonstrate the general effectiveness of our approach across diverse knowledge distillation (KD) paradigms.
>
> To address this concern, we emphasize that our method is **framework-agnostic** and designed to be **plug-and-play**, making it compatible with a wide range of KD frameworks. In our experiments, we applied our method on top of several representative and diverse KD frameworks, including DKD [57], MLKD [20] and TAKD [33] in addition to conventional KD [17] and CRD [45], and observed consistent improvements over their baseline performances (see Table 2 and Table 11).
>
> Additionally, the table below shows further plug-and-play results from applying our methods to Review-KD [6], in comparison with TeKAP. Once again, our method achieves consistent improvements, and this suggests that our approach can complement, rather than replace, existing KD methods.
>
>
> $$
> \\begin{array}{lcc}
> \\textbf{Teacher} & \\text{ResNet-32}\\times\\text{4} & \\text{WideResNet-40-2} & \\text{ResNet-32}\\times\\text{4}    \\\\
> \\textbf{Student} & \\text{ResNet-8}\\times\\text{4} & \\text{WideResNet-40-1}  & \\text{WideResNet-16-2} \\\\ \\hline
> \\textcolor{gray}{\\text{Review-KD w/o aug}} & \\textcolor{gray}{\\text{75.63}} & \\textcolor{gray}{\\text{75.09}} & \\textcolor{gray}{\\text{76.11}} \\\\
> \\text{w/ TeKAP} & 75.32 &  75.15  &76.36     \\\\
> \\text{w/ Angular-KD (ours)} & \\mathbf{75.78} & \\mathbf{75.45} & \\mathbf{76.40}  \\\\
> \\end{array}
> $$
>
> We will revise the manuscript to clarify this point and better highlight the broad compatibility and effectiveness of our approach across various KD frameworks.
>
> ---
>
>
> **[W2] Compared multi-teacher methods are not up-to-date.**
>
> We acknowledge that the compared multi-teacher methods (TAKD [33], published in 2023; and DGKD [40], published in 2021) may not appear very recent. However, to the best of our knowledge, they remain the latest and SoTA multi-teacher KD methods available to date. If any newer approaches in this category emerge, we will incorporate them into our revision accordingly.
>
> ---
>
> **[W3] Additional Visualization.**
>
> Thank you for your helpful suggestion.
> We analyzed two visualizations for our augmented views and the original view, compared to those of TeKAP: 1) UMAP embeddings, and 2) the decision boundaries.
> Although we cannot add figures during rebuttal due to the NeurIPS 2025 rebuttal policy, we observe the following: 1) In UMAP, TeKAP's augmented logits substantially overlap with the teacher's logit, whereas our method yields more dispersed views, aligning with TeKAP's lower ensemble diversity scores (Table 16, supplementary); and 2) In decision boundary plots, our augmented views produce clearly more discriminative boundaries, as our higher accuracy of each view than TeKAP's each view (Table 14, supplementary).
>
>
> We will include additional visualizations of UMAP and decision boundaries, along with a detailed implementation process, in the final version, if accepted.
>
> ---
>
> **[Q1] Results on larger-scale models.**
>
> We appreciate the reviewer’s thoughtful recommendation. To assess scalability, in the following table, we conduct experiments with larger teacher and student configurations: 1) a ResNet-110 teacher and a ResNet-32 student, and 2) a ResNet-110 teacher and a ResNet-20 student.
>
>
> $$
> \\begin{array}{lcc}
> \\textbf{Teacher} & \\text{ResNet-110} & \\text{ResNet-110} \\\\
> \\textbf{Student} & \\text{ResNet-32} & \\text{ResNet-20} \\\\ \\hline
> \\textcolor{gray}{\\text{Na\\'ive KD and CRD w/o aug}} & \\textcolor{gray}{\\text{73.75}} & \\textcolor{gray}{\\text{71.56}} \\\\
> \\text{w/ TeKAP} & 73.86 & 72.02     \\\\
> \\text{w/ Angular-KD (ours)} & \\mathbf{74.89} & \\mathbf{72.52}  \\\\
> \\end{array}
> $$
>
> Across all settings, our methods yield consistent improvements and outperform TeKAP (a SoTA knowledge augmentation method), indicating that our approach remains effective at larger model scales. We will include these results in the final version.
>
>
> Furthermore, in Table 14 of the supplementary, integrating our approach into the multi-teacher framework (TAKD [33]), which has high complexity due to the multi-teacher network, improves distillation performance, underscoring the robustness of our method under high-complexity methods.

---

> ### Author Response · Authors · 2025-08-05
>
> Dear Reviewer TLMU,
>
> Thank you once again for your time and effort in reviewing our work.
>
> As the reviewer-author discussion phase is approaching its end, we would like to kindly check if you have any additional questions or points for clarification.
>
> We would be grateful for the opportunity to address any remaining concerns during the discussion period.
>
> Sincerely,
>
> Authors.

---

> ### Author Response · Authors · 2025-08-07
> **A gentle reminder for Reviewer TLMU**
>
> Dear Reviewer TLMU,
>
> We appreciate your constructive comments for helping us to improve our paper in many aspects.
>
> This is a gentle reminder since the reviewer-author discussion window is coming to a close.
>
> We would like to ask if you have any further questions regarding our submission paper so that we can still respond.
>
> Thanks.
>
> Authors.

---

### Official Review · Reviewer_tUi1 · 2025-07-03

**Clarity:** 3
**Significance:** 2
**Originality:** 3
**Rating:** 4
**Confidence:** 4

**Summary:**

This study proposes inter-angle and intra-angle losses to derive high-diversity perspectives from a single teacher model. Both losses operate to train linear layers to expand the diversity of outputs for the same sample in the feature and logit domains. Through this, the teacher model applies transfer learning to guide the student's learning direction by delivering multi-view supervision for the same data.

**Questions:**

1. Why does the proposed method induce better view-augmentation compared to traditional data augmentation techniques?
2. Can the diversity achieved through view-augmentation be quantitatively demonstrated?
3. Which dataset was used to derive the results presented in Table 7?
4. Given that the diversity of multi-views may vary across classes, can this be accounted for in the method?

**Ethical Concerns:**

["NO or VERY MINOR ethics concerns only"]

**Final Justification:**

Thank you for your thorough and considerate responses to the reviewer’s comments. I believe this manuscript holds sufficient scholarly merit to be recognized for its research value.

**Limitations:**

1. The novelty of the proposed method is somewhat lacking.
2. Additional analytical experiments are insufficient. Comparisons of representation variants based on diversity margins or the number of views ($N$) are needed.
3. The number of training samples in CIFAR-100 should be reviewed for accuracy.
4. The subcaptions (a) and (b) in Table 9 are meaningless and lack context.

**Quality:**

3

**Strengths And Weaknesses:**

$\textbf{Strengths}$
1. The writing is exceptionally clear and well-structured, making the content highly accessible and easy to follow.
2. The approach expands the student model's representation coverage by generating multi-view (akin to negative views) from a single teacher model for the same sample.
3. By pushing the similarity between feature and vector spaces, the method secures flexibility in diversity expansion.
4. The mathematical approach of the proposed method is clearly presented and well-organized for easy comprehension.

$\textbf{Weaknesses}$
1. The novelty of the proposed method is somewhat limited, as it applies the concept of negative pairs from existing contrastive learning to multi-representations of the same sample.
2. The accuracy improvement of the proposed method is not convincingly superior compared to existing methods.

---

> ### Author Rebuttal · Authors · 2025-07-28
>
> Dear reviewer tUi1,
>
> Thank you for your time and efforts in reviewing our manuscript. We appreciate your recognition of our paper’s strengths, particularly its clear writing, well-structured mathematical approach. Below, we would like to provide clarifications and address the concerns and questions you raised.
>
> ---
>
> **[W1, L1] Limited Novelty on mimicking the concept of negative pairs from contrastive learning.**
>
> We would like to clarify that the novelty of our approach lies in promoting a broadened coverage of augmented views via intra- and inter-angular diversity maximization. Although the inter-angle loss, particularly the constraint term in Eq. (3), may superficially resemble conventional contrastive learning, our core formulation, especially the diversity term, is fundamentally different. Rather than treating each view as a negative pair and pulling it indiscriminately from the teacher view, we encourage angular separation among all pairs of multi-views, while preserving their alignment with the teacher. This promotes diversity without sacrificing representational consistency. Intra-angle diversity loss further ensures an even distribution across views around the teacher output. We believe that this fundamental distinction sets our formulation and ideas apart from conventional contrastive learning objectives, which focus on negative separation.
>
> Moreover, our work presents two novel findings: 1) Angularly diversified knowledge from a single teacher can serve as, and even outperform multi-teacher networks (Table 5), and 2) Higher angular diversity theoretically and empirically improves distillation effectiveness by increasing ensemble diversity and lowering ensemble loss (Tables 15 and 16).
>
> To the best of our knowledge, these analyses on the effect of angular diversity are not covered in existing knowledge distillation and ensemble learning studies, underscoring the value of our contributions.
>
>
> ---
>
> **[W2] Minimal accuracy improvement.**
>
> We believe that our improvements are substantial and meaningful, particularly given the strength of the underlying baselines. To better quantify this, we compare the average improvements of our method and the current SoTA augmentation method, TeKAP, relative to the non-augmented baselines across all 18 settings in Table 1. As shown in the “Average Gains” column of the following table, our augmentation method yields an average gain of +1.02pp, whereas TeKAP achieves only +0.36pp from baseline methods.
>
>
> | Method          | Average Gains |
> | --------------- |:------------:|
> | TeKAP      |       +0.36   |
> | Ours           |     **+1.02**    |
>
>
> Our average gain is 2.86 times larger than TeKAP (+1.02pp vs. +0.36pp). We believe this consistent performance gain across diverse settings demonstrates the practical effectiveness of our approach.
>
> ---
>
> **[Q1] Why is our view augmentation better than the data augmentation?**
>
> Thanks for your insightful question. We believe that our view-augmentation provides two major benefits over the data augmentation.
> 1) Efficiency (1 forward pass vs. N forward passes); our method produces multi-views with a single forward pass, whereas data augmentation typically requires N forward passes for N augmented inputs, leading to significantly higher computational cost.
> 2) Evenly dispersed representations; by explicitly maximizing the angular diversity in the representation space, our method ensures that augmented views are evenly dispersed and exhibit minimal redundancy.  In contrast, heuristic data augmentation influences embeddings only indirectly, often resulting in redundant representations or ones that drift away from the teacher’s original view, making it difficult to achieve consistent and balanced dispersion in the representation space.
>
> To further investigate this, we applied standard data augmentations (random flip, color jitter, and random rotation) to generate multiple views and ensemble them for student distillation, similar to our view augmentations.
> The table below shows the CIFAR-100 accuracy for our view-augmented logits vs. the data-augmented logits, where a ResNet-32x4 teacher and a ResNet-8x4 student are used.
>
>
> | Method          | Top-1 Accuracy |
> | --------------- |:------------:|
> | Baseline     |       75.46   |
> | w/ Data augmentation     |     66.91     |
> | w/ View augmentation (ours)           |     **76.46**    |
>
> We observe a substantial performance drop when multiple views are generated via data augmentation.
> To analyze this degradation, we check the accuracy of each augmented view in the table below.
>
> |  View index | 0 | 1  | 2  | 3 | 4 | Teacher acc
> | --------------- |:------------:|  :------------:|  :------------:|  :------------:|  :------------:| :------------:|
> | Data augmentation     |     74.52     |   74.62 | 74.56 | 74.41 |  74.30 | 79.42 |
> | View augmentation (ours)           |     79.23    | 79.30 | 79.42 | 79.34  | 79.19 | 79.42 |
>
> Each augmented view by data augmentation shows significantly lower accuracy due to the noise caused by the data perturbation, leading the overall performance to decline.
>
> We believe that this highlights the effectiveness of our approach in producing consistent and diverse representations without computational overhead.
> We will include these results and a detailed comparison with the data‑augmentation approach in the final version if accepted.
>
>
> ---
>
> **[Q2] How can the diversity introduced by view augmentation be measured quantitatively?**
>
> Yes, we quantify diversity using the ensemble-diversity metric [35]. As shown in the supplementary materials (Table 16), our method achieves 11.633, whereas TeKAP obtains 6.650, demonstrating a substantially higher degree of diversity.
> Table 6 further shows that each of our proposed loss functions contributes to increasing this diversity metric.
> Finally, in Figure 4 of the supplementary, we visualize the correlation matrix (i.e., similarity matrix) between augmented views compared to TeKAP; our approach yields significantly lower similarity between views than TeKAP, further validating that our methods produce more diverse augmented logits.
>
>
> ---
>
> **[Q3] Which dataset was used for Table 7?**
>
> Thank you for catching this omission.
> Table 7 presents results on CIFAR-100.
> We will update the manuscript to indicate this clearly.
>
> ---
>
> **[Q4] Can our method control the diversity at the class level?**
>
> Thank you for your valuable suggestion. Yes. We do so as follows:
>
> First, we compute per-class variance to quantify each class's diversity by measuring the variance over all samples of each class in the table below. As you mentioned, we observe that the diversity varies across classes.
>
> | Class index | 0 | 1 | 2 | ... | 98 | 99 |
> | --------------- |:------------:|  :------------:|  :------------:|  :------------:|  :------------:| :------------:|
> | Variance      |    86.96      |   126.83   | 214.36 | ... | 190.20 | 212.10 |
>
>
> Then, we normalize these variances via the mean value over those of all classes to obtain class-wise weights. Finally, we apply these weights to the inter-angle and intra-angle diversity losses. Formally,
>
>
> $$L^{aug}=\frac{1}{C}\sum_{c=1}^{C}w_c(L^{inter}_c + L^{intra}_c) + L^{GT},$$
>
> where C is the number of classes, L^{inter}\_{c} is the constrained inter-angle diversity loss on the sample x with class c, L^{intra}\_{c} is the intra-angle diversity loss on the sample x with class c, and L^{GT} is the ground-truth loss.
> With this implementation, classes with larger diversity receive larger loss weights, further promoting their views to spread (reducing overlapping).
> The table below reports the performance of this class-wise re-weighting on CIFAR-100.
>
> | Method | Top-1 Accuracy |
> | --------------- |:------------:|
> | KD + CRD w/o aug | 75.46 |
> |w/ Class-wise re-weighting | 76.04 |
> | w/ Ours | 76.46 |
>
>
> We observe that this yields no significant improvement in performance.
> However, since we have confirmed that diversity indeed varies across classes, we will leverage this insight in future work to explore its potential.
>
> ---
>
> **[L2] Additional analytical experiments based on diversity margins or the number of views.**
>
> Thank you for the suggestion. As noted in the main submission, we already report Top-1 accuracy ablations for the angular margin (Table 9b) and the number of views (Table 7), but we agree that further analysis would be valuable for our paper.
> Below, we present extended ablations over additional values of angular margin and the number of views.
>
> | # Views | 1 | 2| 3| 4| 5 | 6| 7 | 8 | 9 |
> | --------------- |:------------:|:------------:|:------------:|:------------:|:------------:|:------------:|:------------:| :------------:| :------------:|
> | Top-1 accuracy  | 75.87 | 75.85  | 76.25 | 76.44 | **76.46** | 76.37 | 76.26 | 76.20 | 76.10 |
>
> | Angular Margin 𝛾 | 0.1 | 0.15 | 0.2  | 0.25 | 0.3 |
> | --------------- |:------------:| :------------:| :------------:| :------------:| :------------:|
> | Top-1 accuracy | 76.34 | 76.29 | **76.46** | 76.20 | 76.31 |
>
> We will include these additional results in the final version, if accepted.
>
> ---
>
> **[L3] Review of CIFAR-100 training sample count.**
>
> As described in Section 4.1 (line 210), CIFAR-100 contains 50,000 training and 10,000 validation images across 100 classes. We will revise the manuscript to ensure this is more clearly stated where CIFAR-100 is mentioned.
>
>
> ---
>
> **[L4] Meaningless subsections (a) and (b) in Table 9.**
>
> Thank you for pointing this out. We will update subsections in Table 9 to: “(a) Effect of diversity and constraint terms” and “(b) Effect of angular margin 𝛾” to ensure each subtable is clearly presented.

---

> ### Author Response · Authors · 2025-08-05
>
> Dear Reviewer tUi1,
>
> Thank you once again for your time and effort in reviewing our work.
>
> As the reviewer-author discussion phase is approaching its end, we would like to kindly check if you have any additional questions or points for clarification.
>
> We would be grateful for the opportunity to address any remaining concerns during the discussion period.
>
> Sincerely,
>
> Authors.

---

> ### Author Response · Authors · 2025-08-07
> **A gentle reminder for Reviewer tUi1**
>
> Dear Reviewer tUi1,
>
> We appreciate your constructive comments for helping us to improve our paper in many aspects.
>
> This is a gentle reminder since the reviewer-author discussion window is coming to a close.
>
> We would like to ask if you have any further questions regarding our submission paper so that we can still respond.
>
> Thanks.
>
> Authors.

---

> ### Comment · Reviewer_tUi1 · 2025-08-09
>
> I would like to express my sincere gratitude to the reviewer for their thoughtful and dedicated responses.
> It is truly encouraging to see that the author not only addressed every question in detail but also took the extra step of conducting new experiments to support their feedback.
> I am glad and appreciative of the effort and care reflected in these responses.
> At the same time, I believe that several of the responses to specific questions should be further refined in the manuscript to ensure a meaningful improvement in the overall quality of the paper.
> I will raise my rating score.

---

### Comment · Area_Chair_dKqS · 2025-08-07
**discussions**

Dear reviewers,

Please take a look at the rebuttals and acknowledge if the authors have addressed (some of) your concerns. And further engage with authors if needed.

---

### Note · Authors · 2025-08-13

Dear Reviewers,

We sincerely thank all reviewers for their dedicated efforts and insightful feedback throughout the review process.

We are grateful that the reviewers acknowledged the key strengths of our paper, including our novel approach (TLMU, SW6m, KCJt), the supporting theoretical analysis (tUi1, TLMU, KCJt), its broad compatibility with existing methods (TLMU, SW6m), and a well-structured presentation (tUi1, SW6m, KCJt).

In the rebuttal, we've tried our best to address the primary concerns raised.
Specifically: (1) we further clarified the novelty of our approach over the concept of the contrastive learning (in response to tUi1), (2) we demonstrated its compatibility with other advanced KD methods via additional plug-and-play experiments (raised by TLMU, KCJt), and (3) we provided extra experiments on binary segmentation to show its general applicability (requested by SW6m).

We sincerely appreciate the reviewers’ feedback, which has strengthened the manuscript, and we hope that our Angular-KD, by examining the effect of angular diversity, can provide valuable insights into the field of knowledge distillation.

Thank you very much,

Authors.

---

### Decision · Program_Chairs · 2025-09-17

**Decision:**

Accept (poster)

**Comment:**

This paper presents a new knowledge distillation method, which uses multiple cheap augmentation views of the teacher for student to learn from. Compared with the most relevant work, the views are learned instead of created with random perturbation. The method also uses additional loss terms to enforce the views to be diverse. Experiments are conducted on the CIFAR-100, STL-10 and TinyImageNet datasets using various teacher-student network pairs, and modest improvements are shown over previous multi-teacher approaches. The authors shall incorporate the reviewers comments in the final version.